# An Adaptive Algorithm for Learning with Unknown Distribution Drift

**Alessio Mazzetto**
Brown University

**Eli Upfal**
Brown University

## Abstract

We develop and analyze a general technique for learning with an unknown distribution drift. Given a sequence of independent observations from the last $T$ steps of a drifting distribution, our algorithm agnostically learns a family of functions with respect to the current distribution at time $T$. Unlike previous work, our technique does not require prior knowledge about the magnitude of the drift. Instead, the algorithm adapts to the sample data. Without explicitly estimating the drift, the algorithm learns a family of functions with almost the same error as a learning algorithm that knows the magnitude of the drift in advance. Furthermore, since our algorithm adapts to the data, it can guarantee a better learning error than an algorithm that relies on loose bounds on the drift. We demonstrate the application of our technique in two fundamental learning scenarios: binary classification and linear regression.

## 1 Introduction

Standard statistical learning models (such as PAC learning) assume independent and identically distributed training set, and evaluate the performance of their algorithms with respect to the same distribution as the training set [Vapnik, 1998, van de Geer, 2000, Shalev-Shwartz and Ben-David, 2014, Wainwright, 2019]. However, in many practical applications, such as weather forecast, finance prediction or consumer preference analysis, the training data is drawn from a non-stationary distribution that drifts in time. In this work, we consider a more general setting where the samples are still independent, but their distribution can change over time. To obtain accurate results, the learning algorithm needs to adjust to the distribution drift occurring in the input.

This framework has been extensively studied in the literature [Helmbold and Long, 1991, Bartlett, 1992, Helmbold and Long, 1994, Barve and Long, 1996, 1997]. This line of research culminated in showing that as long as the total variation distance of two consecutive distributions is bounded by $\Delta$, there exists an algorithm that agnostically learns a family of binary classifiers with VC dimension $\nu$ with expected error $O((\nu\Delta)^{1/3})$ [Long, 1998], which can be shown to be tight. These results were generalized in the work of Mohri and Muñoz Medina [2012] to address any family of functions with bounded Rademacher complexity and to use a finer problem-dependent distance between distributions called discrepancy, originally introduced in the context of domain adaptation [Mansour et al., 2009].

The core idea of the aforementioned work is to learn by using a number of previous samples that minimizes the trade-off between the error due to the variance of the estimation (*statistical error*), and the error due to the drifting of the samples with respect to the current distribution (*drift error*). If the algorithm trains using only a few recent observations, the statistical error will be large. If the algorithm uses a larger training set, including not very recent observations, the drift error will be large. For example, if the algorithm uses the most recent $r$ training point, the hypothesis class has VC-dimension $\nu$, and the distribution drift in each step is bounded by $\Delta$, then the statistical error is $O(\sqrt{\nu/r})$ and the drift error is $O(r\Delta)$. The trade-off with respect to $r$ is optimized for $r = \Theta((\Delta^2/\nu)^{-1/3})$ giving $O((\nu\Delta)^{1/3})$ error, as mentioned before. This, as well as similar approaches

in the literature, requires an upper bound to the magnitude of the drift and resolves the trade-off between the statistical and drift errors based on this knowledge. As noted in previous work [Hanneke and Yang, 2019], it is an open problem to develop an algorithm that adapts to the training set and does not rely on prior knowledge about the drift, whose solution would lead to the practical applicability of those ideas.

Our work resolves this open problem. Our algorithm does not require any prior knowledge of the magnitude of the drift, and it adapts based on the input data. Without explicitly estimating the drift (which is often impossible), the algorithm agnostically learns a family of functions with the same error guarantee as an algorithm that knows the exact magnitude of the drift in advance. Our approach has two advantages: it eliminates the, often unrealistic, requirement of having a bound on the drift, and it gives better results when the drift bounds are not tight. We showcase our algorithm in two important learning settings: binary classification and linear regression.

## 2 Preliminary

Let $(\mathcal{Z}, \mathcal{A})$ be a measurable space. Let $Z_1, \ldots, Z_T$ be a sequence of mutually independent random variables on $\mathcal{Z}$ distributed according to $P_1, \ldots, P_T$ respectively, i.e. $Z_t \sim P_t$ for $t \leq T$. For $r \leq T$, we denote by $P_T^r$ the average distribution of the most recent $r$ distributions $P_{T-r+1}, \ldots, P_T$:

$$P_T^r(A) \doteq \frac{1}{r} \sum_{t=T-r+1}^{T} P_t(A) \qquad \forall A \in \mathcal{A} .$$

We set $\mathbb{P}_T^r$ to be the corresponding empirical distribution over the random variables $Z_{T-r+1}, \ldots, Z_T$:

$$\mathbb{P}_T^r(A) \doteq \frac{|\{Z_t \in A : T - r + 1 \leq t \leq T\}|}{r} \qquad \forall A \in \mathcal{A} .$$

Let $\mathcal{F}$ be a class of measurable functions from $\mathcal{Z}$ to $\mathbb{R}$. For $f \in \mathcal{F}$, and any distributions $P, Q$ on $(\mathcal{Z}, \mathcal{A})$, we let

$$P(f) \doteq \mathbb{E}_{Z \sim P} f(Z) = \int f dP, \qquad \|P - Q\|_{\mathcal{F}} \doteq \sup_{f \in \mathcal{F}} |P(f) - Q(f)| .$$

The norm $\|\cdot\|_{\mathcal{F}}$ is a notion of discrepancy introduced by the work of Mohri and Muñoz Medina [2012] to quantify the error due to the distribution shift with respect to a family of functions $\mathcal{F}$, and it is based on previous work on domain adaptation [Mansour et al., 2009].

The goal is to estimate $P_T(f)$ for all $f \in \mathcal{F}$ using the random variables $Z_1, \ldots, Z_T$. Let $1 \leq r \leq T$, and suppose that we do this estimate by considering the empirical values induced by the most recent $r$ random variables. Then, by using triangle inequality, we have the following decomposition

$$\mathbb{E}\|P_T - \mathbb{P}_T^r\|_{\mathcal{F}} \leq \mathbb{E}\underbrace{\|P_T^r - \mathbb{P}_T^r\|_{\mathcal{F}}}_{\text{statistical error}} + \underbrace{\|P_T - P_T^r\|_{\mathcal{F}}}_{\text{drift error}} . \tag{1}$$

The first term of the upper bound is the expected *statistical error* of the estimation, and quantifies how accurately the empirical values $\{\mathbb{P}_T^r(f) : f \in \mathcal{F}\}$ approximate the expectation of each function $f$ according to $P_T^r$. The second term $\|P_T - P_T^r\|_{\mathcal{F}}$ of the upper bound represents the *drift error*. Intuitively, the statistical error decreases by considering more samples, whereas the drift error can potentially increase since we are considering distributions that are further away from our current distribution. We are looking for the value of $r$ that balances this trade-off between statistical error and drift error.

### 2.1 Statistical Error

Since our results revolve around learning a class of functions $\mathcal{F}$, we first need to assume that $\mathcal{F}$ is "learnable", i.e. the statistical error can be bounded as a function of $r$. For concreteness, we use the following standard assumption that the family of $\mathcal{F}$ satisfies the standard machine learning uniform convergence requirement with rate $O(1/\sqrt{r})$.

**Assumption 1.** *(Uniform Convergence). There exists non-negative constants $C_{\mathcal{F},1}$ and $C_{\mathcal{F},2}$ such that for any fixed $r \leq T$ and $\delta \in (0, 1)$, it holds*

$$\mathbb{E}\|P_T^r - \mathbb{P}_T^r\|_{\mathcal{F}} \leq \frac{C_{\mathcal{F},1}}{\sqrt{r}} ,$$

*and with probability at least $1 - \delta$, we have*

$$\|P_T^r - \mathbb{P}_T^r\|_{\mathcal{F}} \leq \frac{C_{\mathcal{F},1}}{\sqrt{r}} + C_{\mathcal{F},2}\sqrt{\frac{\ln(1/\delta)}{r}} \ .$$

The learnability of a family of functions $\mathcal{F}$ is an extensively studied topic in the statistical learning literature (e.g., [Bousquet et al., 2003, Wainwright, 2019]). For a family of binary functions, the above assumption is equivalent to $\mathcal{F}$ having a finite VC-dimension, in which case $C_{\mathcal{F},1} = O(\nu)$ and $C_{\mathcal{F},2} = O(1)$. For a general family of functions $\mathcal{F}$, a sufficient requirement is that the Rademacher complexity of the first $r$ samples is $O(r^{-1/2})$ and the range of any function in $\mathcal{F}$ is uniformly bounded. There is nothing special about the rate $O(r^{-1/2})$. It is possible to adapt our analysis to any rate $O(r^{-\alpha})$ with $\alpha \in (0, 1)$ by modifying the constants of our algorithm.

## 2.2 Quantifying the Drift Error

While for many classes $\mathcal{F}$, it is possible to provide an upper bound to the statistical error by using standard statistical learning theory tools, the drift error is unknown and challenging to estimate. The literature used different approaches to quantify the drift error. By using triangle inequality, it is possible to show that for any $r \leq T$, we have the following upper bounds to the drift error:

$$\|P_T - P_T^r\|_{\mathcal{F}} \leq \frac{1}{r}\sum_{t=1}^{r}\|P_T - P_{T-t}\|_{\mathcal{F}} \leq \max_{t<r}\|P_T - P_{T-t}\|_{\mathcal{F}} \leq \sum_{t=1}^{r-1}\|P_{T-t} - P_{T-t+1}\|_{\mathcal{F}} \ . \quad (2)$$

In a long line of research (e.g., [Bartlett, 1992, Long, 1998, Mohri and Muñoz Medina, 2012, Hanneke and Yang, 2019]), it is assumed that an upper bound to the drift error is known apriori. One of the most used assumption is that there exists a known upper bound $\Delta$ to the discrepancy between any two consecutive distributions, in which case we can upper bound $\sum_{t=1}^{r-1}\|P_{T-t} - P_{T-t+1}\|_{\mathcal{F}}$ with $r\Delta$. In this case, for a binary family $\mathcal{F}$ with VC-dimension $\nu$, we obtain that:

$$\mathbb{E}\|P_T - P_T^r\|_{\mathcal{F}} \lesssim \sqrt{\frac{\nu}{r}} + r\Delta \ ,$$

and we can choose the value of $r$ that minimizes this upper bound. Since these algorithms rely on an unrealistic assumption that an upper bound to the drift is known a priori, they are not usable in practice. It is an open problem to provide a competitive algorithm that can choose $r$ adaptively and it is oblivious to the magnitude of the drift.

Another sequence of work [Mohri and Muñoz Medina, 2012, Awasthi et al., 2023] relaxes the problem setting assuming that the algorithm can observe multiple samples from each distribution $P_1, \ldots, P_T$. In this case, they can provably estimate the discrepancies between different distributions, and compute a weighting of the samples that minimizes a trade-off between the statistical error and the estimated discrepancies. This strategy does not apply to our more general setting, as we can have access to at most one sample from each distribution.

Surprisingly, we show that we can adaptively choose the value of $r$ to minimize the trade-off between statistical error and drift error without explicitly estimating the discrepancy. Noticeably, our method does not require any additional assumption on the drift. The only requirement for our algorithm is that we can compute the norm $\|\cdot\|_{\mathcal{F}}$ from a set of samples. This is formalized as follows.

**Assumption 2.** *(Computability).* *There exists a procedure that computes $\|\mathbb{P}_T^r - \mathbb{P}_T^{2r}\|_{\mathcal{F}}$ for any $r \leq T/2$.*

In general, the hardness of computing the norm $\|\cdot\|_{\mathcal{F}}$ depends on the family $\mathcal{F}$. This challenge is also common in previous work that uses this norm to quantify the distribution drift [Mansour et al., 2009, Awasthi et al., 2023]. In this paper, we provide two examples of important learning settings where this assumption is satisfied: binary classification with zero-one loss and linear regression with squared loss.

## 3 Main Result

Under those assumption, we prove the following theorem, which is our main result.

**Theorem 1.** *Let $\delta \in (0, 1)$. Let Assumptions 1 and 2 hold. Given $Z_1, \ldots, Z_T$, Algorithm 1 outputs a value $\hat{r} \leq T$ such that with probability at least $1 - \delta$, it holds that*

$$\|P_T - \mathbb{P}_T^{\hat{r}}\|_{\mathcal{F}} = O\left(\min_{r \leq T}\left[\frac{C_{\mathcal{F},1}}{\sqrt{r}} + \max_{t < r}\|P_T - P_{T-t}\|_{\mathcal{F}} + C_{\mathcal{F},2}\sqrt{\frac{\log(\log(r+1)/\delta)}{r}}\right]\right)$$

In order to appreciate this theorem, we can observe the following. If we learn using the most recent $r$ samples, similarly to (1) we have the following error decomposition

$$\|P_T - \mathbb{P}_T^r\|_{\mathcal{F}} \leq \|P_T^r - \mathbb{P}_T^r\|_{\mathcal{F}} + \|P_T - P_T^r\|_{\mathcal{F}}$$

By using Assumption 1 and (2), we have that with probability at least $1 - \delta$ it holds that

$$\|P_T - \mathbb{P}_T^r\|_{\mathcal{F}} \leq \frac{C_{\mathcal{F},1}}{\sqrt{r}} + C_{\mathcal{F},2}\sqrt{\frac{\log(1/\delta)}{r}} + \max_{t < r}\|P_T - P_{T-t}\|_{\mathcal{F}} . \tag{3}$$

Theorem 1 guarantees a learning error that is essentially within a multiplicative constant factor as good as the upper bound obtained by selecting the optimal choice of $r$ in (3). This result provides an affirmative answer to the open problem posed by Hanneke and Yang [2019], that asked if it was possible to adaptively choose the value of $r$ that minimizes (3) [1]. The upper bound of the theorem contains a negligible additional factor $\log(r+1)$ within the logarithm due to the union bound required to consider multiple candidate values of $r$. As further evidence of the efficiency of our algorithm, assume that there is no drift, and we are in the usual i.i.d. setting where $P_1 = \ldots = P_T$. In this setting, the following corollary immediately follows from Theorem 1 by setting the drift error equal to 0.

**Corollary 2.** *Consider the setting of Theorem 1, and also assume that $P_1 = \ldots = P_T$. With probability at least $1 - \delta$, the window size $\hat{r}$ computed by Algorithm 1 satisfies*

$$\|P_T - \mathbb{P}_T^{\hat{r}}\|_{\mathcal{F}} = O\left(\frac{C_{\mathcal{F},1}}{\sqrt{T}} + C_{\mathcal{F},2}\sqrt{\frac{\log(\log(T+1)/\delta)}{T}}\right)$$

Observe that in the i.i.d. case, Assumption 1 implies that with probability at least $1 - \delta$, it holds $\|P_T - \mathbb{P}_T^T\|_{\mathcal{F}} \leq C_{\mathcal{F},1}/\sqrt{T} + C_{\mathcal{F},2}\sqrt{\log(1/\delta)/T}$. Corollary 2 shows that with our algorithm we obtain a result that is competitive except for a negligible extra factor $O(\log T)$ within the logarithm. We want to emphasize that our algorithm does not know in advance whether the data is i.i.d. or drifting, and this factor is the small cost of our algorithm to adapt between those two cases.

## 4 Algorithm

We describe an algorithm that attains the results of Theorem 1. Throughout the remaining of this section, we let Assumptions 1 hold. In particular, the algorithm has access to constants $C_{\mathcal{F},1}$ and $C_{\mathcal{F},2}$ that satisfy this assumption. The main challenge is the fact that the drift error $\|P_T - \mathbb{P}_T^r\|_{\mathcal{F}}$ is unknown for any $r > 1$, and it is challenging to quantify since we have only a single sample $Z_T \sim P_T$.

We first provide an informal description of the algorithm. Our algorithm revolves around the following strategy. We do not try to estimate the drift. Instead, we try to assess whether increasing the sample size can yield a better upper bound on the error. Recall that given the most recent $r$ samples, we have the following upper bound on the error,

$$\mathbb{E}\|P_T - \mathbb{P}_T^r\|_{\mathcal{F}} \leq \frac{C_{\mathcal{F},1}}{\sqrt{r}} + \|P_T - P_T^r\|_{\mathcal{F}}. \tag{4}$$

The algorithm cannot evaluate (4) since the drift component, $\|P_T - P_T^r\|_{\mathcal{F}}$, is unknown. Our key idea is to compare the difference in the upper bound on the error when using the latest $2r$ or $r$ samples:

$$\left(\frac{C_{\mathcal{F},1}}{\sqrt{2r}} - \frac{C_{\mathcal{F},1}}{\sqrt{r}}\right) + \left(\|P_T - P_T^{2r}\|_{\mathcal{F}} - \|P_T - P_T^r\|_{\mathcal{F}}\right) \leq \left(\frac{C_{\mathcal{F},1}}{\sqrt{2r}} - \frac{C_{\mathcal{F},1}}{\sqrt{r}}\right) + \|P_T^r - P_T^{2r}\|_{\mathcal{F}} \tag{5}$$

---

[1] We prove a stronger result. The original formulation of the question uses the looser upper bound $\sum_{t=1}^{r-1}\|P_{T-t} - P_{T-t+1}\|_{\mathcal{F}}$ to the drift error in (3), which can be significantly worse as shown in an example of Section 5

The last step follows from the triangle inequality: $\|P_T - P_T^{2r}\|_{\mathcal{F}} \leq \|P_T - P_T^r\|_{\mathcal{F}} + \|P_T^r - P_T^{2r}\|_{\mathcal{F}}$. By considering the latest $2r$ samples rather than $r$ samples, we can see from (5) that the statistical error decreases, and the difference in drift error can be upper bounded by $\|P_T^{2r} - P_T^r\|_{\mathcal{F}}$. We can estimate $\|P_T^r - P_T^{2r}\|_{\mathcal{F}}$ using $\|\mathbb{P}_T^r - \mathbb{P}_T^{2r}\|_{\mathcal{F}}$ within expected error $O(C_{\mathcal{F},1}/\sqrt{r})$ as it depends on $\geq r$ samples. This suggests the following algorithm. Let $r$ be the current sample size considered by the algorithm. The algorithm starts from $r$ equal to 1, and doubles the sample size as long as $\|\mathbb{P}_T^r - \mathbb{P}_T^{2r}\|_{\mathcal{F}}$ is small. If $\|\mathbb{P}_T^r - \mathbb{P}_T^{2r}\|_{\mathcal{F}}$ is big enough, a substantial drift must have occurred in the distributions $P_{T-2r+1}, \ldots, P_T$, and we can show that this implies that $\max_{t < 2r-1} \|P_T - P_{T-t}\|_{\mathcal{F}}$ is also large. When this happens, we can stop our algorithm and return the current sample size.

For a formal description of the algorithm, let $\delta \in (0,1)$ denote the probability of failure of our algorithm, and let $r_i = 2^i$, for $i \geq 0$, be the size of the training set used by the algorithm at iteration $i + 1$. For ease of notation, we set

$$C_{\mathcal{F},\delta} \doteq C_{\mathcal{F},1} + C_{\mathcal{F},2}\sqrt{2\ln(\pi^2/(6\delta))},$$

and

$$S(r,\delta) \doteq \frac{C_{\mathcal{F},\delta}}{\sqrt{r}} + C_{\mathcal{F},2}\sqrt{\frac{2\ln(\log_2(r) + 10)}{r}}. \tag{6}$$

The proofs of the following propositions appear in the Appendix A.

**Proposition 3.** *With probability at least $1 - \delta$, the following event holds:*

$$\|P_T^{r_i} - \mathbb{P}_T^{r_i}\|_{\mathcal{F}} \leq S(r_i, \delta) \qquad \forall i \geq 0$$

We assume that the event of Proposition 3 holds, otherwise our algorithm fails (with probability $\leq \delta$). Our algorithm considers the following *inflated* upper bound $U(r,\delta)$ to $\|\mathbb{P}_T^r - P_T\|_{\mathcal{F}}$ defined as follows

$$U(r,\delta) \doteq 21 \cdot S(r,\delta) + \|P_T - P_T^r\|_{\mathcal{F}}. \tag{7}$$

Proposition 3 implies that with probability at least $1 - \delta$ for any $i \geq 0$, we have

$$U(r_i,\delta) = 21 \cdot S(r_i,\delta) + \|P_T - P_T^{r_i}\|_{\mathcal{F}} \geq \|\mathbb{P}_T^{r_i} - P_T^{r_i}\|_{\mathcal{F}} + \|P_T - P_T^{r_i}\|_{\mathcal{F}} \geq \|P_T - \mathbb{P}_T^{r_i}\|_{\mathcal{F}}$$

We use this value as an upper bound that our algorithm guarantees if we choose a sample size $r_i$. We observe that with respect to (4), the upper bound of the algorithm also contains an additional term that is proportional to $\sqrt{\log(1/\delta)}$ for the high-probability guarantee, and a term proportional to $\sqrt{\log\log r_i}$ that is necessary for the union bound across all possible window sizes $r_i$ for $i \geq 0$. This union bound is required as we need to have a correct estimation for all possible sample sizes $r_i$ in order to assure that the algorithm takes a correct decision at each step. The constant factor in front of the upper bound on the statistical error $S(r_i,\delta)$ is a technical detail that allows taking into account the error of the estimation of the difference in drift error.

We want to compare the upper bound $U(r_i,\delta)$ with the upper bound $U(r_{i+1},\delta)$ obtained by doubling the current sample size $r_i$. As we previously discussed, it is possible to show that if $\|\mathbb{P}_T^{r_i} - \mathbb{P}_T^{r_{i+1}}\|_{\mathcal{F}}$ is sufficiently small, then $U(r_{i+1},\delta) \leq U(r_i,\delta)$, and this intuition is formalized in the following proposition.

**Proposition 4.** *Assume that the event of Proposition 3 holds and let $i \geq 0$.*

$$\text{If } \|\mathbb{P}_T^{r_i} - \mathbb{P}_T^{r_{i+1}}\|_{\mathcal{F}} \leq 4S(r_i,\delta) \text{ then } U(r_{i+1},\delta) \leq U(r_i,\delta).$$

The algorithm works as follows. Starting from $i = 0$, we iteratively increase $i$ by one while $\|\mathbb{P}_T^{r_i} - \mathbb{P}_T^{r_{i+1}}\|_{\mathcal{F}} \leq 4S(r_i,\delta)$. There are two cases. In the first case, we reach $r_i = \lfloor \log_2 T \rfloor$, and this implies that $r_{i+1} > T$. In this case, the algorithm returns $r_i$, and Proposition 4 guarantees that the sample size returned by the algorithm is as good as any previously considered sample size. In the second case, we reach a value of $i$ such that $\|\mathbb{P}_T^{r_i} - \mathbb{P}_T^{r_{i+1}}\|_{\mathcal{F}} > 4S(r_i,\delta)$. In this case, we return $r_i$, and we can still prove that this is a good choice. In fact, as shown in the next proposition, this terminating condition implies a lower bound $\max_{t < r_{i+1}} \|P_T - P_{T-i}\|_{\mathcal{F}}$, thus any estimation with a number of recent samples greater or equal to $r_{i+1}$ could have a non-negligible drift error.

**Algorithm 1:** Adaptive Learning Algorithm under Drift

---

$i \leftarrow 0$ ;
**while** $r_i \leq T/2$ **do**
    **if** $\|\mathbb{P}_T^{r_i} - \mathbb{P}_T^{r_{i+1}}\|_{\mathcal{F}} \leq 4S(r_i, \delta)$ **then**
        $i \leftarrow i + 1$ ;
    **end**
    **else**
        **return** $r_i$ ;
    **end**
**end**
**return** $r_i$

---

**Proposition 5.** *Assume that the event of Proposition 3 holds, and assume that there exists $i \geq 0$ such that $\|\mathbb{P}_T^{r_i} - \mathbb{P}_T^{r_{i+1}}\|_{\mathcal{F}} > 4S(r_i, \delta)$, then $\max_{t < r_{i+1}} \|P_T - P_{T-t}\|_{\mathcal{F}} > S(r_i, \delta)$.*

The pseudo-code of the algorithm is reported in Algorithm 1. The algorithm receives in input $\delta$, the samples $Z_1, \ldots, Z_T$, and returns an integer $\hat{r} \in \{1, \ldots, T\}$ that satisfies Theorem 1.

*Proof of Theorem 1.* We assume that the event of Proposition 3 holds. If it doesn't, we say that our algorithm fails, and this happens with probability $\leq \delta$. Let $\hat{r} = r_j = 2^j$ for $j \geq 0$ be the value returned by the algorithm when it terminates. We remind that our algorithm guarantees an upper bound $U(r_j, \delta) \geq \|P_T - \mathbb{P}_T^{r_j}\|_{\mathcal{F}}$ to the learning error by using $r_j$ samples. Let $r^*$ be the value of $r$ that minimizes this expression

$$r^* = \operatorname*{argmin}_{r \leq T} \left( 21 \cdot S(r, \delta) + \max_{t < r} \|P_T - P_{T-t}\|_{\mathcal{F}} \right) \ ,$$

and let $B^*$ be the minimum value of this expression, i.e.

$$B^* = 21 \cdot S(r^*, \delta) + \max_{t < r^*} \|P_T - P_{T-t}\|_{\mathcal{F}}$$

be any fixed optimal sample size, where we remind the definition of $U$ from (7). The first observation is that the right-hand side of the inequality in Theorem 1 is $O(B^*)$. Therefore, in order to prove the theorem, it is sufficient to show that $U(r_j, \delta)/B^* = O(1)$.

We distinguish two cases: $(a)$ $r^* < 2r_j$, and $(b)$ $r^* \geq 2r_j$. We first consider case $(a)$. We let $k \geq 0$ be the largest integer such that $r_k \leq r^*$. Since $r^* < 2r_j$, it holds that $k \leq j$. Since our algorithm returned $r_j$, Proposition 4 applies for $i = 0, \ldots, j-1$, thus $U(r_j, \delta) \leq U(r_k, \delta)$. We have that:

$$\frac{U(r_j, \delta)}{B^*} \leq \frac{U(r_k, \delta)}{B^*} \tag{8}$$

We observe that

$$\frac{U(r_k, \delta)}{B^*} = \frac{21S(r_k, \delta) + \|P_T - P_T^{r_k}\|_{\mathcal{F}}}{21S(r^*, \delta) + \max_{t < r^*} \|P_T - P_{T-t}\|_{\mathcal{F}}} \leq \frac{21S(r_k, \delta) + \max_{t < r_k} \|P_T - P_{T-t}\|_{\mathcal{F}}}{21S(r^*, \delta) + \max_{t < r^*} \|P_T - P_{T-t}\|_{\mathcal{F}}}$$

$$\leq \frac{S(r_k, \delta)}{S(r^*, \delta)} + \frac{\max_{t < r_k} \|P_T - P_{T-t}\|_{\mathcal{F}}}{\max_{t < r^*} \|P_T - P_{T-t}\|_{\mathcal{F}}} \leq \sqrt{\frac{r^*}{r_k}} + 1 \leq 3 \ ,$$

where the first inequality is due to (2), and the last inequality is due to the fact that $r_k \leq r^* < 2r_k$ by definition of $r_k$. By using the above inequality in (8), we obtain that $U(r_j, \delta)/B^* \leq 3$.

We consider case $(b)$. Since $T \geq r^* \geq 2r_j = r_{j+1}$, the algorithm returned $r_j$ because $\|\mathbb{P}_T^{r_j} - \mathbb{P}_T^{r_{j+1}}\|_{\mathcal{F}} > 4S(r_j, \delta)$. Using Proposition 5, we have

$$\max_{t < r^*} \|P_T - P_{T-t}\|_{\mathcal{F}} \geq \max_{t < r_{j+1}} \|P_T - P_{T-t}\|_{\mathcal{F}} > S(r_j, \delta) \ .$$

Therefore, we have that

$$\frac{U(r_j, \delta)}{B^*} = \frac{21S(r_j, \delta) + \|P_T - P_T^{r_j}\|_{\mathcal{F}}}{21S(r^*, \delta) + \max_{t < r^*} \|P_T - P_{T-t}\|_{\mathcal{F}}}$$

$$\leq \frac{21S(r_j, \delta)}{\max_{t < r^*} \|P_T - P_{T-t}\|_{\mathcal{F}}} + \frac{\max_{t < r_j} \|P_T - P_{T-t}\|_{\mathcal{F}}}{\max_{t < r^*} \|P_T - P_{T-t}\|_{\mathcal{F}}} \leq \frac{21S(r_j, \delta)}{S(r_j, \delta)} + 1 = 21$$

This concludes the proof. $\qquad \square$

# 5 Binary Classification with Distribution Drift

In this section, we show an application of Theorem 1 for the fundamental statistical learning problem of agnostic learning a family of binary classifiers. Let $\mathcal{Z} = \mathcal{X} \times \mathcal{Y}$, where $\mathcal{X}$ is the feature space, and $\mathcal{Y} = \{0, 1\}$ is the label space, i.e. $Z_t = (X_t, Y_t)$. A hypothesis class $\mathcal{H}$ is a class of functions $h : \mathcal{X} \mapsto \mathcal{Y}$ that classify the feature space $\mathcal{X}$. Given a point $(x, y) \in \mathcal{X} \times \mathcal{Y}$ and a function $h \in \mathcal{H}$, the risk of $h$ on $(x, y)$ is defined through the following function $L_h(x, y) = \mathbf{1}_{\{y \neq h(x)\}}$. We work with the class of functions $\mathcal{F} = \{L_h : h \in \mathcal{H}\}$.

Let $h^* = \operatorname{argmin}_{h \in \mathcal{H}} P_T(L_h)$ be a function with minimum expected risk with respect to the current distribution $P_T$. We want to use Theorem 1 to find a function $h \in \mathcal{H}$ such that $P_T(L_h)$ is close to $P_T(L_{h^*})$. Let $\nu$ be the VC dimension of $\mathcal{H}$. The VC dimension describes the complexity of the family $\mathcal{H}$, and it is used to quantify the statistical error. In particular, using standard learning tools, it is possible to show that the family $\mathcal{F}$ satisfies Assumption 1 on the sample complexity with constants $C_{\mathcal{F},1} = O(\sqrt{\nu})$ and $C_{\mathcal{F},2} = O(1)$ (e.g., [Mohri et al., 2018]).

Finally, to satisfy Assumption 2, we need to exhibit a procedure that for $r \leq T/2$, outputs the quantity $\|\mathbb{P}_T^r - \mathbb{P}_T^{2r}\|_{\mathcal{F}}$. This quantity is also referred to as $\mathcal{Y}$-discrepancy between the empirical distributions $\mathbb{P}_T^r$ and $\mathbb{P}_T^{2r}$ in previous work on transfer learning [Mohri and Muñoz Medina, 2012]. We can adapt a strategy from Ben-David et al. [2010] to our setting and show that it is possible to compute it by solving an empirical risk minimization problem. We say that a hypothesis class $\mathcal{H}$ is *computationally tractable* if given a finite set of points from $\mathcal{X} \times \mathcal{Y}$, there exists an algorithm that returns a hypothesis $h$ that achieves the minimum risk over this set of points.

**Lemma 6.** *Assume that $\mathcal{H}$ is symmetric, i.e. $h \in \mathcal{H} \iff 1 - h \in \mathcal{H}$. For $r \leq T/2$, it holds*

$$\|\mathbb{P}_T^r - \mathbb{P}_T^{2r}\|_{\mathcal{F}} = \frac{1}{2} - \frac{1}{2} \inf_{h \in \mathcal{H}} \left[ \frac{1}{r} \sum_{t=T-r+1}^{T} L_h(X_t, 1 - Y_t) + \frac{1}{r} \sum_{t=T-2r+1}^{T-r} L_h(X_t, Y_t) \right]$$

Given $r$, the minimum in Lemma 6 can be computed by solving an empirical risk minimization over the most recent $2r$ points, where we flip the label of half of those points, i.e. we use the points $(X_{T-2r+1}, Y_{T-2r+1}), \ldots, (X_{T-r}, Y_{T-r}), (X_{T-r+1}, 1-Y_{T-r+1}), \ldots, (X_T, 1-Y_T)$. Thus, if $\mathcal{H}$ is computationally tractable and symmetric, Assumption 2 holds. This is indeed true for many hypothesis class, e.g., hyperplanes, axis-aligned rectangles, or threshold functions. However, the empirical risk minimization problem could be expensive to solve exactly, but as we discuss in Section 8, it is possible to modify the algorithm to allow for an approximation of $\|\mathbb{P}_T^r - \mathbb{P}_T^{2r}\|_{\mathcal{F}}$.

Our main result for binary classification is given in the following theorem:

**Theorem 7.** *Let $\mathcal{H}$ be a computationally tractable and symmetric binary class with VC dimension $\nu$. Let $\mathcal{F} = \{L_h : h \in \mathcal{H}\}$. Let $\hat{r} \leq T$ be output of Algorithm 1 with input $Z_1, \ldots, Z_T$ using the family $\mathcal{F}$. Let $\hat{h} = \operatorname{argmin}_{h \in \mathcal{H}} \mathbb{P}_T^{\hat{r}}(L_h)$ be an empirical risk minimizer over the most recent $\hat{r}$ samples. With probability at least $1 - \delta$, the following holds:*

$$P_T(L_{\hat{h}}) - P_T(L_{h^*}) = O\left( \min_{r \leq T} \left[ \sqrt{\frac{\nu}{r}} + \max_{t < r} \|P_T - P_{T-t}\|_{\mathcal{F}} + \sqrt{\frac{\log(\log(r+1)/\delta)}{r}} \right] \right)$$

The symmetry assumption is not necessary, and in the appendix we show how to remove it at the cost of a more expensive computation of the discrepancy. It is instructive to compare this upper bound with the results of previous work. An often used assumption in the literature, originally introduced in Bartlett [1992], is that there is a known value $\Delta > 0$, such that drift in each step is bounded by $\Delta$, i.e. for all $t < T$, $\|P_{t+1} - P_t\|_{\mathcal{F}} \leq \Delta$. Assume that $T$ is sufficiently large, i.e. $T = \Omega((\nu/\Delta^2)^{1/3})$. By using this assumption on the drift, previous work showed that with high-probability, they can find a classifier $\hat{h}$ such that $P_T(L_{\hat{h}}) - P_T(L_{h^*}) = O\left(\sqrt[3]{\Delta\nu}\right)$, and it can be shown that this upper bound is tight up to constants within those assumptions [Barve and Long, 1996]. These previous works assumed they had access a priori to the value of $\Delta$, since those algorithms compute $\hat{h}$ by solving a empirical risk minimization over a number of previous samples $\Theta((\nu/\Delta^2)^{1/3})$ that is decided before observing the data. On the other hand, this assumption on the drift together with (2) implies that $\max_{t < r} \|P_T - P_{T-t}\|_{\mathcal{F}} \leq (r-1)\Delta$. Our algorithm (Theorem 7) guarantees an error that depends on a minimum choice over $r \leq T$, hence it is always smaller than

the one obtained with a specific choice of $r$. If we choose $r = (\nu/\Delta^2)^{1/3}$ in the upper bound of Theorem 7, we can show that with high-probability, our algorithm returns a classifier $\hat{h}$ such that $P_T(L_{\hat{h}}) - P_T(L_{h^*}) = O\left(\sqrt[3]{\Delta\nu} + \sqrt[6]{\Delta^2/\nu}\sqrt{\log\log(\nu/\Delta^2)}\right)$. Our algorithm achieves this guarantee while being adaptive with respect to the upper bound $\Delta$, and it can indeed guarantee a better result when this upper bound is loose.

For example, assume an extreme case in which the algorithm is given a $\Delta > 0$ bound on the drift in each step, but the training set has actually no drift, it was all drawn from the distribution $P_T$. Previous methods are oblivious to the actual data, and they guarantee an upper bound $O\left(\sqrt[3]{\Delta\nu}\right)$ with high-probability, since they decide the sample size $\Theta((\nu/\Delta^2)^{1/3})$ a priory without observing the input. In contrast, our algorithm adapts to this scenario, and Theorem 7 guarantees that we obtain an $O\left(\sqrt{\nu/T} + \sqrt{(\log\log T)/T}\right)$ error, with high-probability, essentially retrieving the error guarantee for learning with independent and identically distributed samples. Observe that our upper bound depends on $T$, and it goes to 0 when $T \to \infty$.

It is possible to show that our algorithm can obtain asymptotically better guarantee even if $\|P_t - P_{t-1}\|_{\mathcal{F}} = \Delta \ll 1$ for all $t < T$, i.e. there is an exact drift of $\Delta$ at each step. This is because our algorithm provides a guarantee as a function of $\max_{t<r}\|P_T - P_{T-t}\|_{\mathcal{F}}$ rather than the looser quantity $\sum_{t=1}^{r-1}\|P_{T-t} - P_{T-t+1}\|_{\mathcal{F}}$, as shown in the following example. Let $\mathcal{X} = [0,1]$, and let $\mathcal{H}$ be a class of threshold functions over $[0,1]$ i.e. for any $c \in [0,1]$, there exists classifiers $h_c, h'_c \in \mathcal{H}$ such that $h_c(x) = 1$ if and only if $x \geq c$ and $h'_c(x) = 1$ if and only if $x < c$. The class $\mathcal{H}$ has VC-dimension equal to 2. We construct a sequence of distributions $P_1, \ldots, P_T$ as follows. The marginal distribution over $\mathcal{X}$ is uniform for each distribution $P_t$ with $t = 1, \ldots, T$. At time $t \leq T$, the classification of $x \in [0,1]$ is given by a function $\ell_t : [0,1] \mapsto \{0,1\}$. Assume that there exists two disjoint intervals $I_0 = [p_0, p_0 + \Delta/2)$ and $I_1 = [p_1, p_1 + \Delta/2)$ such that $\ell_T(x) = 0$ for all $x \in I_0$ and $\ell_T(x) = 1$ for all $x \in I_1$. We let

$$\ell_{T-t}(x) = \begin{cases} 1 - \ell_T(x) & \text{if } (x \in I_0) \wedge (t \text{ is odd}) \\ 1 - \ell_T(x) & \text{if } (x \in I_1) \wedge (t \text{ is even}) \\ \ell_T(x) & \text{otherwise} \end{cases} , \qquad \forall x \in \mathcal{X}, t < T .$$

In particular, $\ell_{T-t}(x)$ differs from $\ell_T(x)$ for $x \in I_0$ if $t$ is odd, and for $x \in I_1$ if $t$ is even. By construction, $\|P_t - P_{t-1}\|_{\mathcal{F}} = \Delta$ for all $t < T$, i.e. there is an exact drift of $\Delta$ at each step (to be precise, in the last step $\|P_T - P_{T-1}\|_{\mathcal{F}} = \Delta/2$). As discussed before, with the assumption of a bounded drift $\Delta$ at each step, previous methods guarantee an upper bound $O(\sqrt[3]{\Delta})$ with high-probability. However, we also have that for any $1 \leq r \leq T$, it holds by construction that

$$\max_{t<r}\|P_T - P_{T-t}\|_{\mathcal{F}} \leq \Delta .$$

Hence, our algorithm (Theorem 7) guarantees with high-probability an upper bound

$$O\left(\Delta + \sqrt{\nu/T} + \sqrt{(\log\log T)/T}\right) .$$

Since our algorithm is adaptive with respect to the drift, it can correctly use more samples. In contrast, previous non-adaptive algorithms that rely on the assumption of bounded drift $\Delta$ at each step choose a sample size of $\Theta(\Delta^{-2/3})$ based on this assumption, thus they can only guarantee a looser bound of $O(\Delta^{1/3})$, even when this assumption is satisfied with equality.

It is possible to use the recent lower bound strategy of Mazzetto and Upfal [2023] in order to show that the upper bound of Theorem 7 is essentially tight in a minimax sense.

**Theorem 8.** *Let $\mathcal{H}$ be a binary class with VC dimension $\nu$, and consider an arbitrary non-decreasing sequence $\Delta_1 = 0, \Delta_2, \ldots, \Delta_T$ of non-negative real numbers. Let $\mathcal{F} = \{L_h : h \in \mathcal{H}\}$. Let $\mathcal{A}$ be any algorithm that observes $Z_1, \ldots, Z_T$, and it outputs a classifier $h_{\mathcal{A}} \in \mathcal{H}$. If*

$$\Phi^* = \min_{r \leq T}\left(\sqrt{\frac{\nu}{r}} + \Delta_r\right) < 1/3 ,$$

*then, for any algorithm $\mathcal{A}$, there exists a sequence of distributions $P_1, \ldots, P_T$ such that $\max_{t<r}\|P_T - P_{T-t}\|_{\mathcal{F}} \leq \Delta_r$ for any $r \leq T$, and with probability at least $1/8$ it holds that:*

$$P_T(L_{h_{\mathcal{A}}}) - P_T(L_{h^*}) = \Omega(\Phi^*) = \Omega\left(\min_{r \leq T}\left(\sqrt{\frac{\nu}{r}} + \max_{t<r}\|P_T - P_{T-t}\|_{\mathcal{F}}\right)\right)$$

# 6 Linear Regression with Squared Loss

In the previous section, we showed an application of Theorem 1 for the problem of binary classification with zero-one loss. In this section, we show that our main result can also be applied to a linear regression problem. Similarly to the previous section, we let $\mathcal{Z} = \mathcal{X} \times \mathcal{Y}$, where $\mathcal{X}$ is the feature space and $\mathcal{Y} = [-1, 1]$ is the label space, i.e. $Z_t = (X_t, Y_t)$. In this section, we constrain the feature space $\mathcal{X} = \{x \in \mathbb{R}^d : \|x\|_2 \leq 1\}$ to be the unit ball centered at the origin in $\mathbb{R}^d$.

We consider the $\ell_2$ regularized linear prediction class $\mathcal{H} = \{x \mapsto \langle x, w \rangle : w \in \mathbb{R}^d \wedge \|w\|_2 \leq 1\}$. We denote each predictor $h_w \in \mathcal{H}$ with its weight $w$. To evaluate the quality of a prediction $y = h_w(x)$, we use the squared loss $L : \mathbb{R} \times \mathbb{R} \mapsto \mathbb{R}$ defined as $L(y, y') = (y - y')^2$. For each $h_w \in \mathcal{H}$, we let $L_w(x, y) = L(h_w(x), y)$ be the function that evaluates the loss incurred by $h_w$ for any $z = (x, y) \in \mathcal{Z}$. We work with the family of functions $\mathcal{F} = \{L_w : h_w \in \mathcal{H}\}$. By a standard uniform convergence argument based on the Rademacher complexity of $\mathcal{F}$ (e.g., see [Kakade et al., 2008, Shamir, 2015, Awasthi et al., 2020]), we have that for any $1 \leq r \leq T$, it holds:

$$\|P_T^r - \mathbb{P}_T^r\|_{\mathcal{F}} = O\left(\frac{1}{\sqrt{r}} + \sqrt{\frac{\ln(1/\delta)}{r}}\right) ,$$

thus we satisfy Assumption 1 with $C_{\mathcal{F},1} = O(1)$ and $C_{\mathcal{F},2} = O(1)$.

The computation of the discrepancy is more challenging. Let $1 \leq r \leq T/2$. Using the definition of $\mathcal{F}$, we have that:

$$\|\mathbb{P}_T^r - \mathbb{P}_T^{2r}\|_{\mathcal{F}} = \frac{1}{2r} \sup_{w:\|w\|\leq 1} \left| \sum_{t=T-r+1}^{T} \left(Y_t - \langle X_t, w \rangle\right)^2 - \sum_{t=T-2r+1}^{T-r} \left(Y_t - \langle X_t, w \rangle\right)^2 \right| . \quad (9)$$

Let $c_r \in \mathbb{R}$, $b_r \in \mathbb{R}^d$, and $A_r \in \mathbb{R}^{d \times d}$ be defined as follows:

$$a_r = \sum_{t=T-r+1}^{T} Y_t^2 - \sum_{t=T-2r+1}^{T-r} Y_t^2, \qquad b_r = \sum_{t=T-2r+1}^{T-r} Y_t X_t - \sum_{t=T-r+1}^{T} Y_t X_t ,$$

$$A_r = \sum_{t=T-r+1}^{T} X_t^T X_t - \sum_{t=T-2r+1}^{T-r} X_t^T X_t ,$$

and observe that the matrix $A_r$ is symmetric. Using those definition, we can manipulate the right-hand side of (9) to show that it is equivalent to:

$$\max\left(-a_r + \inf_{w:\|w\|\leq 1} \left[w^T(-A_r)w - 2b_r^T w\right], a_r + \inf_{w:\|w\|\leq 1} \left[w^T(A_r)w - 2(-b_r^T)w\right]\right) \quad (10)$$

In order to compute (10), it is sufficient to be able to solve the minimization problem

$$\inf_{w:\|w\|\leq 1} \left(w^T A w - 2b^T w\right) \quad (11)$$

where $A \in \mathbb{R}^{d \times d}$ is a symmetric matrix, and $b \in \mathbb{R}^d$. This minimization problem has been extensively studied for the trust-region method [Conn et al., 2000], and Hager [2001, Section 2] provides a way to compute the solution of (11) in terms of a diagonalization of $A$. Thus, we also satisfy Assumption 2, and we can obtain the following result as an immediate corollary of Theorem 1.

**Theorem 9.** *Let $\delta \in (0, 1)$. Let $1 \leq \hat{r} \leq T$ be the output of Algorithm 1 with input $Z_1, \ldots, Z_T$ and using the family $\mathcal{F}$ described in this section. Let $\hat{w} = \arg\min_{w:\|w\|\leq 1} \mathbb{P}_T^{\hat{r}}(L_w)$ be the linear classifier with minimum loss over the most recent $\hat{r}$ samples. Then, with probability at least $1 - \delta$:*

$$P_T(L_{\hat{w}}) - P_T(L_{w^*}) = O\left(\min_{r \leq T} \left[\frac{1}{\sqrt{r}} + \max_{t<r}\|P_T - P_{T-t}\|_{\mathcal{F}} + \sqrt{\frac{\log(\log(r+1)/\delta)}{r}}\right]\right)$$

*where $w^* = \arg\min_{w:\|w\|\leq 1} P_T(L_w)$ is the linear predictor with minimum loss with respect to the current distribution $P_T$.*

# 7 Related Work

Additional variants of learning with distribution drift have been studied in the literature. Freund and Mansour [1997] provide a refined learning algorithm in the special case of rapid distribution shift with a constant direction of change. In the work of Bartlett et al. [2000], they show specialized bounds in the case of infrequent changes and other different restrictions on the distribution drift. The work of Crammer et al. [2010] provides regret bound for online learning with an adversarial bounded drift. Yang [2011] studies the problem of active learning in a distribution drift setting. Hanneke et al. [2015] provide an efficient polynomial time algorithm to learn a class of linear separators with a drifting target concept under the uniform distribution in the realizable setting. Interestingly, they also show how to adapt their algorithm with respect to an unknown drift, although their technique relies on the realizability of the learning problem. In the more recent work of Hanneke and Yang [2019], they relax the independence assumption and provide learning guarantees for a sequence of random variables that is both drifting and mixing.

# 8 Conclusion, Limitations and Future Directions

We present a general learning algorithm that adapts to an unknown distribution drift in the training set. Unlike previous work, our technique does not require prior knowledge about the magnitude of the drift. For the problem of binary classification, we show that without explicitly estimating the drift, there exists an algorithm that learns a binary classifier with the same or better error bounds compared to the state-of-the-art results that rely on prior information about the magnitude of the drift. This is a major step toward practical solutions to the problem since prior knowledge about the distribution drift in the training set is often hard to obtain.

We presented concrete results for binary classification and linear regression, but our technique can be applied for learning any family of functions $\mathcal{F}$ as long as it is possible to compute the distance $\|\mathbb{P}_T^r - \mathbb{P}_T^{2r}\|_{\mathcal{F}}$ between the empirical distributions with $r$ and $2r$ samples according to the $\|\cdot\|_{\mathcal{F}}$ norm (Assumption 2). This is often a challenging problem, and it is related to the computation of the discrepancy between distributions, which was studied in previous work on transfer learning [Mansour et al., 2009, Ben-David et al., 2010]. For binary classification, we assume that the empirical risk minimization problem is tractable. However, an exact solution to this problem is computationally hard for many hypothesis classes of interest, and this is a limitation of our algorithm and previous work on transfer learning. In those cases, we can modify our analysis to use the best-known approximation for the distance $\|\mathbb{P}_T^r - \mathbb{P}_T^{2r}\|_{\mathcal{F}}$ as long as there is an approximation guarantee with respect to its exact value. As an illustrative example, if we have a procedure that returns an approximation $E_r$ such that $1 \le \|\mathbb{P}_T^r - \mathbb{P}_T^{2r}\|_{\mathcal{F}}/E_r \le \alpha$ for any $r \ge 1$, then it is possible to change the algorithm to obtain a guarantee that is at the most a factor $O(\alpha^2)$ worse than the one achieved by Algorithm 1 with the exact computation of the distance (we refer to Appendix B for additional details).

The method presented here uses a distribution-independent upper bound for the statistical error. While this upper bound can be tight in the worst-case, as shown in our lower bound for binary classification (Theorem 8), it can be loose for some other sequence of distributions. As a future direction, it is an interesting problem to provide an adaptive algorithm with respect to the drift that uses distribution-dependent upper bounds, for example, based on the Rademacher complexity, which can be possibly computed from the input data. Our algorithm does not naturally extend to this setting, as it requires knowing the rate at which the upper bound on the statistical error is decreasing (see proof of Proposition 4).

**Acknowledgements.** This material is based on research sponsored by Defense Advanced Research Projects Agency (DARPA) and Air Force Research Laboratory (AFRL) under agreement number FA8750-19-2-1006 and by the National Science Foundation (NSF) under award IIS-1813444. The U.S. Government is authorized to reproduce and distribute reprints for Governmental purposes notwithstanding any copyright notation thereon. The views and conclusions contained herein are those of the authors and should not be interpreted as necessarily representing the official policies or endorsements, either expressed or implied, of Defense Advanced Research Projects Agency (DARPA) and Air Force Research Laboratory (AFRL) or the U.S. Government.

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

# A Deferred Proofs

*Proof of Proposition 3.* For any $i \geq 0$, let $\delta_i = 6\delta/[\pi^2(i+10)^2]$. By using Assumption 1, we have that with probability at least $1 - \delta_i$ it holds that

$$\|P_T^{r_i} - \mathbb{P}_T^{r_i}\|_{\mathcal{F}} \leq \frac{C_{\mathcal{F},1} + C_{\mathcal{F},2}\sqrt{\ln(\pi^2/6) + \ln(1/\delta) + 2\ln(i+10)}}{\sqrt{r_i}}$$

$$\leq \frac{C_{\mathcal{F},1} + \sqrt{2}C_{\mathcal{F},2}\sqrt{\ln(\pi^2/(6\delta)) + \ln(i+10)}}{\sqrt{r_i}}$$

$$\leq \frac{C_{\mathcal{F},\delta}}{\sqrt{r_i}} + C_{\mathcal{F},2}\sqrt{\frac{2\ln(i+10)}{r_i}}$$

where in the second inequality we used the definition of $C_{\mathcal{F},\delta}$ and the fact that $\sqrt{x+y} \leq \sqrt{x} + \sqrt{y}$. Additionaly, the following equality holds

$$\sum_{i=0}^{\infty} \delta_i \leq \delta\left(\frac{6}{\pi^2}\right)\sum_{i=1}^{\infty} 1/i^2 = \delta ,$$

where in the last equality, we used the known fact that $\sum_{i=1}^{\infty} 1/i^2 = \pi^2/6$. Thus, if we take an union bound over all $i \geq 0$, we have with probability at least $1 - \delta$ it holds that

$$\|P_T^{r_i} - \mathbb{P}_T^{r_i}\|_{\mathcal{F}} \leq \frac{C_{\mathcal{F},\delta}}{\sqrt{r_i}} + C_{\mathcal{F},2}\sqrt{\frac{2\ln(i+10)}{r_i}} \qquad \forall i \geq 0 .$$

The statement follows by observing that the right-hand side of the above inequality is equal to $S(r_i, \delta)$ for any $i \geq 0$. $\qquad \square$

*Proof of Proposition 4.* . We have

$$U(r_{i+1}, \delta) - U(r_i, \delta) = 21[S(r_{i+1}, \delta) - S(r_i, \delta)] + \|P_T - P_T^{r_{i+1}}\|_{\mathcal{F}} - \|P_T - P_T^{r_i}\|_{\mathcal{F}} .$$

By using the triangle inequality, we obtain

$$\|P_T - P_T^{r_{i+1}}\|_{\mathcal{F}} \leq \|P_T - P_T^{r_i}\|_{\mathcal{F}} + \|P_T^{r_i} - P_T^{r_{i+1}}\|_{\mathcal{F}} ,$$

thus we have

$$U(r_{i+1}, \delta) - U(r_i, \delta) \leq 21[S(r_{i+1}, \delta) - S(r_i, \delta)] + \|P_T^{r_i} - P_T^{r_{i+1}}\|_{\mathcal{F}} . \tag{12}$$

We use the triangle inequality and Proposition 3 to show that

$$\|P_T^{r_i} - P_T^{r_{i+1}}\|_{\mathcal{F}} \leq \|P_T^{r_i} - \mathbb{P}_T^{r_i}\|_{\mathcal{F}} + \|P_T^{r_{i+1}} - \mathbb{P}_T^{r_{i+1}}\|_{\mathcal{F}} + \|\mathbb{P}_T^{r_i} - \mathbb{P}_T^{r_{i+1}}\|_{\mathcal{F}}$$

$$\leq \|\mathbb{P}_T^{r_i} - \mathbb{P}_T^{r_{i+1}}\|_{\mathcal{F}} + S(r_i, \delta) + S(r_{i+1}, \delta) .$$

If we plug the above inequality in (12) and use the assumption that $\|\mathbb{P}_T^{r_i} - \mathbb{P}_T^{r_{i+1}}\|_{\mathcal{F}} \leq 4S(r_i, \delta)$, we obtain

$$U(r_{i+1}, \delta) - U(r_i, \delta) \leq 21[S(r_{i+1}, \delta) - S(r_i, \delta)] + S(r_{i+1}, \delta) + 5S(r_i, \delta)$$

$$= 22S(r_{i+1}, \delta) - 16S(r_i, \delta)$$

If we expand the above upper bound by using the definition of the function $S$, we have

$$22S(r_{i+1}, \delta) - 16S(r_i, \delta) = \frac{C_{\mathcal{F},\delta}}{\sqrt{r_{i+1}}}[22 - 16\sqrt{2}] + C_{\mathcal{F},2}\sqrt{\frac{2\ln(i+11)}{r_{i+1}}}\left[22 - 16\sqrt{2}\sqrt{\frac{\ln(i+10)}{\ln(i+11)}}\right]$$

$$\leq 0 ,$$

where the last inequality follows since $22 - 16\sqrt{2} \leq 0$, and $22 - 16\sqrt{2}\sqrt{\ln(i+10)/\ln(i+11)} \leq 0$. Thus, it holds that

$$U(r_{i+1}, \delta) - U(r_i, \delta) \leq 22S(r_{i+1}, \delta) - 16S(r_i, \delta) \leq 0 ,$$

and we can conclude that $U(r_{i+1}, \delta) \leq U(r_i, \delta)$. $\qquad \square$

*Proof of Proposition 5.* . Observe that by using the triangle inequality, it holds that

$$\|\mathbb{P}_T^{r_i} - \mathbb{P}_T^{r_{i+1}}\|_{\mathcal{F}} \leq \|P_T^{r_i} - \mathbb{P}_T^{r_i}\|_{\mathcal{F}} + \|P_T^{r_{i+1}} - \mathbb{P}_T^{r_{i+1}}\|_{\mathcal{F}} + \|P_T^{r_i} - P_T^{r_{i+1}}\|_{\mathcal{F}}$$
$$\leq \|P_T^{r_i} - P_T^{r_{i+1}}\|_{\mathcal{F}} + S(r_i, \delta) + S(r_{i+1}, \delta)$$
$$\leq \|P_T^{r_i} - P_T^{r_{i+1}}\|_{\mathcal{F}} + 2S(r_i, \delta) \quad,$$

where in the second inequality we used Proposition 3. By assumption, we have that $\|\mathbb{P}_T^{r_i} - \mathbb{P}_T^{r_{i+1}}\|_{\mathcal{F}} \geq 4S(r_{i+1}, \delta)$, hence

$$\|P_T^{r_i} - P_T^{r_{i+1}}\|_{\mathcal{F}} \geq 2S(r_i, \delta) \tag{13}$$

Observe that by triangle inequality, we have that

$$\|P_T^{r_i} - P_T^{r_{i+1}}\|_{\mathcal{F}} \leq \|P_T - P_T^{r_i}\|_{\mathcal{F}} + \|P_T - P_T^{r_{i+1}}\|_{\mathcal{F}} \quad.$$

We use (2) and obtain that

$$\|P_T^{r_i} - P_T^{r_{i+1}}\|_{\mathcal{F}} \leq \max_{t < r_i}\|P_T - P_{T-t}\|_{\mathcal{F}} + \max_{t < r_{i+1}}\|P_T - P_{T-t}\|_{\mathcal{F}}$$
$$\leq 2\max_{t < r_{i+1}}\|P_T - P_{T-t}\|_{\mathcal{F}} \quad.$$

By combining the above inequality with (13), we finally obtain that $\max_{t < r_{i+1}}\|P_T - P_{T-t}\|_{\mathcal{F}} \geq S(r_i, \delta)$. $\qquad\square$

*Proof of Lemma 6.* We remind that $\mathcal{F} = \{L_h : h \in \mathcal{H}\}$. By using the definition of $\|\cdot\|_{\mathcal{F}}$, we have

$$\|\mathbb{P}_T^r - \mathbb{P}_T^{2r}\|_{\mathcal{F}} = \sup_{h \in \mathcal{H}}\left|\frac{1}{r}\sum_{t=T-r+1}^{T} L_h(X_t, Y_t) - \frac{1}{2r}\sum_{t=T-2r+1}^{T} L_h(X_t, Y_t)\right|$$
$$= \frac{1}{2}\sup_{h \in \mathcal{H}}\left|\frac{1}{r}\sum_{t=T-r+1}^{T} L_h(X_t, Y_t) - \frac{1}{r}\sum_{t=T-2r+1}^{T-r} L_h(X_t, Y_t)\right| \quad.$$

We can remove the absolute value by re-writing this expression as

$$\|\mathbb{P}_T^r - \mathbb{P}_T^{2r}\|_{\mathcal{F}} = \frac{1}{2}\max\left\{\sup_{h \in \mathcal{H}}\left(\frac{1}{r}\sum_{t=T-r+1}^{T} L_h(X_t, Y_t) - \frac{1}{r}\sum_{t=T-2r+1}^{T-r} L_h(X_t, Y_t)\right),\right.$$
$$\left.\sup_{h \in \mathcal{H}}\left(\frac{1}{r}\sum_{t=T-2r+1}^{T-r} L_h(X_t, Y_t) - \frac{1}{r}\sum_{t=T-r+1}^{T} L_h(X_t, Y_t)\right)\right\} \quad. \tag{14}$$

Consider the first argument of the above maximum. We can observe that for any $(X, Y) \in \mathcal{X} \times \mathcal{Y}$, it holds that $L_h(X, Y) + L_h(X, 1 - Y) = 1$, thus we obtain that

$$\sup_{h \in \mathcal{H}}\left(\frac{1}{r}\sum_{t=T-r+1}^{T} L_h(X_t, Y_t) - \frac{1}{r}\sum_{t=T-2r+1}^{T-r} L_h(X_t, Y_t)\right)$$
$$= \sup_{h \in \mathcal{H}}\left(1 - \frac{1}{r}\sum_{t=T-r+1}^{T} L_h(X_t, 1 - Y_t) - \frac{1}{r}\sum_{t=T-2r+1}^{T-r} L_h(X_t, Y_t)\right)$$
$$= 1 - \inf_{h \in \mathcal{H}}\left(\frac{1}{r}\sum_{t=T-r+1}^{T} L_h(X_t, 1 - Y_t) + \frac{1}{r}\sum_{t=T-2r+1}^{T-r} L_h(X_t, Y_t)\right) \tag{15}$$

Similarly, we can demonstrate that the second term of the maximum is equal to:

$$\sup_{h \in \mathcal{H}}\left(\frac{1}{r}\sum_{t=T-2r+1}^{T-r} L_h(X_t, Y_t) - \frac{1}{r}\sum_{t=T-r+1}^{T} L_h(X_t, Y_t)\right)$$
$$= 1 - \inf_{h \in \mathcal{H}}\left(\frac{1}{r}\sum_{t=T-r+1}^{T} L_h(X_t, Y_t) + \frac{1}{r}\sum_{t=T-2r+1}^{T-r} L_h(X_t, 1 - Y_t)\right) \quad. \tag{16}$$

Therefore, to compute the discrepancy (14), it is sufficient to solve the two empirical risk minimization problems in (15) and (16). To obtain the final statement, we can observe that for any $(X, Y)$ and $h \in \mathcal{H}$, it holds that $L_h(X, Y) = L_{1-h}(X, 1 - Y)$. Thus, we can show that (16) is equivalent to:

$$1 - \inf_{h \in \mathcal{H}} \left( \frac{1}{r} \sum_{t=T-r+1}^{T} L_h(X_t, Y_t) + \frac{1}{r} \sum_{t=T-2r+1}^{T-r} L_h(X_t, 1 - Y_t) \right)$$

$$= 1 - \inf_{h \in \mathcal{H}} \left( \frac{1}{r} \sum_{t=T-r+1}^{T} L_{1-h}(X_t, 1 - Y_t) + \frac{1}{r} \sum_{t=T-2r+1}^{T-r} L_{1-h}(X_t, Y_t) \right) \ .$$

Since $\mathcal{H}$ is symmetric, i.e. $1 - h \in \mathcal{H} \iff h \in \mathcal{H}$, we can conclude that (15) and (16) have the same value. This concludes the proof.

$\square$

*Proof of Theorem 7.* Since $\mathcal{H}$ has VC-dimension $\nu$, by a standard argument we have that the family $\mathcal{F} = \{L_h : h \in \mathcal{H}\}$ has VC-dimension upper bounded by $2\nu$, thus it satisfies Assumption 1 on the sample complexity for uniform convergence with $C_{\mathcal{F},1} = O(\sqrt{\nu})$ and $C_{\mathcal{F},2} = O(1)$.

We can observe that since $\mathcal{H}$ is symmetric, Lemma 6 shows that we can compute $\|\mathbb{P}_T^r - \mathbb{P}_T^{2r}\|_{\mathcal{F}}$ for any $r \geq 1$ by solving an empirical risk minimization problem. Since $\mathcal{H}$ is computationally tractable, there exists a procedure that solves this problem, thus we also satisfy Assumption 2.

**Remark:** If the symmetry assumption does not hold, in the proof of Lemma 6 we show that we can still compute the discrepancy $\|\mathbb{P}_T^r - \mathbb{P}_T^{2r}\|_{\mathcal{F}}$ by solving the two empirical risk minimization problems in (15) and (16).

Hence, we can use Algorithm 1 with the family $\mathcal{F}$, and let $\hat{r}$ be the value returned by the algorithm. Theorem 1 guarantees that with probability at least $1 - \delta$, we have that

$$\|P_T - \mathbb{P}_T^{\hat{r}}\|_{\mathcal{F}} = O \left( \min_{r \leq T} \left[ \sqrt{\frac{\nu}{r}} + \max_{t < r} \|P_T - P_{T-t}\|_{\mathcal{F}} + \sqrt{\frac{\log(\log(r+1)/\delta)}{r}} \right] \right) \ . \quad (17)$$

Now, we have that

$$
\begin{aligned}
P_T(L_{\hat{h}}) - P_T(L_{h^*}) = P_T(L_{\hat{h}}) - P_T(L_{h^*}) &= P_T(L_{\hat{h}}) - \mathbb{P}_T^{\hat{r}}(L_{\hat{h}}) + \mathbb{P}_T^{\hat{r}}(L_{\hat{h}}) - P_T(L_{h^*}) \\
&\leq P_T(L_{\hat{h}}) - \mathbb{P}_T^{\hat{r}}(L_{\hat{h}}) + \mathbb{P}_T^{\hat{r}}(L_{h^*}) - P_T(L_{h^*}) \\
&\leq |P_T(L_{\hat{h}}) - \mathbb{P}_T^{\hat{r}}(L_{\hat{h}})| + |\mathbb{P}_T^{\hat{r}}(L_{h^*}) - P_T(L_{h^*})| \\
&\leq 2\|P_T - \mathbb{P}_T^{\hat{r}}\|_{\mathcal{F}},
\end{aligned}
$$

where the first inequality is due to the definition of $\hat{h}$. Therefore, using (17), we have that with probability at least $1 - \delta$, it holds

$$P_T(L_{\hat{h}}) - P_T(L_{h^*}) = O \left( \min_{r \leq T} \left[ \sqrt{\frac{\nu}{r}} + \max_{t < r} \|P_T - P_{T-t}\|_{\mathcal{F}} + \sqrt{\frac{\log(\log(r+1)/\delta)}{r}} \right] \right) \ .$$

$\square$

*Proof of Theorem 9.* The theorem is proven following the same structure as the proof of Theorem 7.

$\square$

### A.1 Lower Bound.

In this section, we prove the lower bound of Theorem 8. The proof structure is based on the work of Mazzetto and Upfal [2023]. We provide a simpler statement of their proof in our setting, and we remove the additional regularity assumption used in that work to characterize the drift error. We introduce the following notation.

We say that a distribution $Q$ over $\mathcal{Z}^n$ is a product distribution if it can be written as the product of $n$ distributions over $\mathcal{Z}$, i.e. $Q = Q_1 \times \ldots \times Q_n$. For any $n \geq 1$, we can observe that since the random

variables $Z_1, \ldots, Z_n$ are independent, their distribution can be described as a product distribution over $\mathcal{Z}^n$. Given two strings $\tau, \tau' \in \{-1, 1\}^n$, we let $\mathrm{hd}(\tau, \tau') = \frac{1}{2}\|\tau - \tau'\|_1$ be the Hamming distance between the two strings, i.e. the number of positions in which the two strings differ.

Let $\nu$ be the VC dimension of the hypothesis class $\mathcal{H}$. We will construct a (later defined) family of product distributions $\mathcal{Q} = \{Q^{(\tau)} = Q_1^{(\tau)} \times \ldots \times Q_T^{(\tau)} : \tau \in \{-1, 1\}^\nu\}$ over $\mathcal{Z}^T$ that are indexed by a string $\tau \in \{-1, 1\}^\nu$. Intuitively, each distribution $\mathcal{Q}^{(\tau)}$ is a possible candidate for the distribution of the random variables $Z_1, \ldots, Z_T$. We will show that for any algorithm $\mathcal{A}$, there exists a product distribution $Q^{(\tau)}$ such that the classifier $h_\mathcal{A}$ computed by $\mathcal{A}$ using the samples $Z_1, \ldots, Z_T$ from $Q^{(\tau)}$ has large expected error. The proof is based on Assouad's Lemma. We provide a statement of this lemma that is an adaptation of its classical statement to our setting [Yu, 1997].

**Lemma 10** (Assouad's Lemma). *Let $\mathcal{Q}$ be defined as above. For any function $g : \mathcal{Z}^T \mapsto \{-1, 1\}^\nu$, there exists $\tau \in \{-1, 1\}^\nu$ such that*

$$\mathbb{E}_{Z \sim Q^{(\tau)}} \mathrm{hd}\big(g(Z_1, \ldots, Z_n), \tau\big) \geq \frac{\nu}{2} \cdot \min_{\substack{\tau', \tau'' \\ \mathrm{hd}(\tau', \tau'')=1}} \|Q^{(\tau')} \wedge Q^{(\tau'')}\|_1$$

Let $\Delta_1 = 0, \Delta_2, \ldots, \Delta_T$ be the sequence defined in the statement of Theorem 8. We let $\Phi : \{1, \ldots, T\} \mapsto \mathbb{R}$ be the function

$$\Phi(r) = \left( \sqrt{\frac{\nu}{r}} + \Delta_r \right) \ .$$

We let $\Phi^* = \min_r \Phi(r)$, and we remind that we assume $\Phi^* < 1/3$ in the statement of the Theorem. We build the family $\mathcal{Q}$ based on the following value:

$$\tilde{r} = \max \left\{ 1 \leq r \leq T : \Delta_r < \sqrt{\frac{\nu}{r}} \right\}$$

**Proposition 11.** *The following holds:*

$$\Phi(\tilde{r}) \leq 3\Phi^* \ .$$

*Proof.* Let $r^* \in \{1, \ldots, r\}$ be a value such that $\Phi^* = \Phi(r^*)$. The statement follows by exploiting the definition of $\tilde{r}$. We distinguish two cases. If $\tilde{r} \geq r^*$, we have that

$$\frac{\Phi(\tilde{r})}{\Phi^*} = \frac{\Delta_{\tilde{r}} + \sqrt{\nu/\tilde{r}}}{\Delta_{r^*} + \sqrt{\nu/r^*}} \leq \frac{2\sqrt{\nu/\tilde{r}}}{\sqrt{\nu/r^*}} = 2\sqrt{r^*/\tilde{r}} \leq 2$$

Conversely, if $r^* > \tilde{r}$, we have that

$$\frac{\Phi(\tilde{r})}{\Phi^*} = \frac{\Delta_{\tilde{r}} + \sqrt{\nu/\tilde{r}}}{\Delta_{r^*} + \sqrt{\nu/r^*}} = \frac{\Delta_{\tilde{r}} + \sqrt{\nu/(\tilde{r}+1)}\sqrt{(\tilde{r}+1)/\tilde{r}}}{\Delta_{r^*} + \sqrt{\nu/r^*}} \leq 3\Delta_{\tilde{r}+1}/\Delta_{r^*} \leq 3$$

In the first inequality we used the fact that the sequence $\Delta_1, \ldots, \Delta_T$ is non-decreasing, and the inequality $\sqrt{\nu/(\tilde{r}+1)} \leq \Delta_{\tilde{r}+1}$ due to the definition ot $\tilde{r}$. $\square$

We define the family of product distributions $\mathcal{Q}$ as follows. Let $\overline{X}_1, \ldots, \overline{X}_\nu$ be a shatter set for the hypothesis class $\mathcal{H}$. We build the following family $\mathcal{Q} = \{Q^{(\tau)} = Q_1^{[\tau]} \times \ldots \times Q_T^{[\tau]} : \tau \in \{-1, 1\}^\nu\}$ of product distributions over $\mathcal{Z}^T = (\mathcal{X} \times \mathcal{Y})^T$ that are indexed by $\tau \in \{-1, 1\}^\nu$. They are defined as follows:

$$\Pr_{(X,Y) \sim Q_t^{(\tau)}} (Y = 1 | X = \overline{X}_i) = \begin{cases} \frac{1}{2} + \frac{\tau_i}{16\sqrt{6}} \left( \sqrt{\frac{\nu}{\tilde{r}}} + \Delta_{\tilde{r}} - \Delta_{T-t+1} \right) & \text{if } t > T - \tilde{r} \\ \frac{1}{2} & \text{else} \end{cases}$$

$$\Pr_{(X,Y) \sim Q_t^{(\tau)}} (X = \overline{X}_i) = \frac{1}{\nu} \qquad\qquad \forall i \in \{1, \ldots, \nu\}$$

Those distributions are well-defined. In fact, we have that for all $t > T - \tilde{r}$:

$$\frac{1}{16\sqrt{6}}\left(\sqrt{\frac{\nu}{\tilde{r}}} + \Delta_{\tilde{r}} - \Delta_{T-t+1}\right) \leq \frac{1}{16\sqrt{6}}\left(\sqrt{\frac{\nu}{\tilde{r}}} + \Delta_{\tilde{r}}\right) \leq \frac{1}{16\sqrt{6}}\Phi(\tilde{r}) \leq \frac{3}{10}\Phi^* < 1/4 \ ,$$

where we used Proposition 11. Given a classifier $h \in \mathcal{H}$ and $\tau \in \{-1, 1\}^\nu$, we remind that

$$Q_T^{(\tau)}(L_h) = \Pr_{(X,Y)\sim Q_T^{(\tau)}}(h(X) \neq Y) \ .$$

We can observe that for any classifier $h \in \mathcal{H}$ and $t > T - \tilde{r}$, it holds by construction that:

$$\left|Q_T^{(\tau)}(L_h) - Q_{T-t+1}^{(\tau)}(L_h)\right| = \frac{1}{16\sqrt{6}}\Delta_{T-t+1} \leq \Delta_{T-t+1} \ ,$$

and for any $t \leq T - \tilde{r}$, we have that

$$\left|Q_T^{(\tau)}(L_h) - Q_{T-t}^{(\tau)}(L_h)\right| = \frac{1}{16\sqrt{6}}\left(\Delta_{\tilde{r}} + \sqrt{\frac{\nu}{\tilde{r}}}\right) \leq \frac{1}{8\sqrt{6}}\sqrt{\frac{\nu}{\tilde{r}}} = \frac{1}{8\sqrt{6}}\sqrt{\frac{\nu}{\tilde{r}+1}}\sqrt{\frac{\tilde{r}+1}{\tilde{r}}}$$

$$\leq \frac{1}{8\sqrt{3}}\Delta_{\tilde{r}+1} \leq \Delta_{T-t+1} \ ,$$

where the first and the second inequality are due to the definition of $\tilde{r}$, and the last inequality follows from the fact that the sequence $\Delta_1, \ldots, \Delta_T$ is non-decreasing. Hence, if we let $\mathcal{F} = \{L_h : h \in \mathcal{H}\}$, it results that for $\tau$ and for any $1 \leq r \leq T$, it holds that:

$$\max_{t<r}\|Q_T^{(\tau)} - Q_{T-t}^{(\tau)}\|_{\mathcal{F}} \leq \Delta_r \ . \tag{18}$$

We also let $L_{h^*}$ be the minimum loss that is achieved by a function $h \in \mathcal{H}$ with respect to $Q_T^{(\tau)}$, i.e. $L_{h^*}^\tau = \mathrm{argmin}_{L_h:h\in\mathcal{H}} Q_T^{(\tau)}(L_h)$. By using the family $\mathcal{Q}$ together with Assouad's Lemma, we can show the following.

**Lemma 12.** *Let $\mathcal{Q}$ be defined as above. Let $\mathcal{A} : \mathcal{Z}^T \mapsto \mathcal{H}$ be any algorithm that observes a sequence of elements $T$ elements from $\mathcal{Z}$, and it outputs a classifier $h_\mathcal{A}$. For any algorithm $\mathcal{A}$, there exists $\tau \in \{-1, 1\}^\nu$ such that if the input $Z_1, \ldots, Z_n$ is sampled according to $Q^{(\tau)}$, then:*

$$\mathbb{E}\left[Q_T^{(\tau)}(L_{h_\mathcal{A}}) - Q_T^{(\tau)}(L_{h^*}^\tau)\right] \geq \frac{7\sqrt{6}\Phi^*}{768} \ .$$

*Proof.* Let $\tau \in \{-1, 1\}^\nu$. We can define $\tau_\mathcal{A} = \left(2h_\mathcal{A}(\overline{X}_1) - 1, \ldots, 2 \cdot h_\mathcal{A}(\overline{X}_\nu) - 1\right) \in \{-1, 1\}^\nu$. We have that:

$$Q_T^{(\tau)}(L_{h_\mathcal{A}}^\tau) = \left[\frac{1}{2} - \frac{1}{16\sqrt{6}}\left(\sqrt{\frac{\nu}{\tilde{r}}} + \Delta_{\tilde{r}}\right)\right] + \frac{1}{8\sqrt{6}\nu}\left(\sqrt{\frac{\nu}{\tilde{r}}} + \Delta_{\tilde{r}}\right)\sum_{i=1}^\nu \mathbf{1}_{\{\tau_{\mathcal{A},i}\neq\tau_i\}}$$

$$= \left[\frac{1}{2} - \frac{1}{16\sqrt{6}}\left(\sqrt{\frac{\nu}{\tilde{r}}} + \Delta_{\tilde{r}}\right)\right] + \frac{\Phi(\tilde{r})}{8\sqrt{6}\nu}\mathrm{hd}(\tau_\mathcal{A}, \tau)$$

By construction of $Q_T^{(\tau)}$, we can observe that

$$Q_T^{(\tau)}(L_{h^*}^\tau) = \left[\frac{1}{2} - \frac{1}{16\sqrt{6}}\left(\sqrt{\frac{d}{\tilde{r}}} + \Delta_{\tilde{r}}\right)\right] \ ,$$

hence, we have the following relation

$$Q_T^{(\tau)}(L_{h_\mathcal{A}}) - Q_T^{(\tau)}(L_{h^*}^\tau) = \frac{\Phi(\tilde{r})}{8\sqrt{6}\nu}\mathrm{hd}(\tau_\mathcal{A}, \tau) \tag{19}$$

$$\implies \mathbb{E}\left[Q_T^{(\tau)}(L_{h_\mathcal{A}}) - Q_T^{(\tau)}(L_{h^*}^\tau)\right] = \frac{\Phi(\tilde{r})}{8\sqrt{6}\nu}\mathbb{E}\,\mathrm{hd}(\tau_\mathcal{A}, \tau)$$

Observe that $\mathcal{A}$ can be seen as a map from $\mathcal{Z}^T$ to $\{-1,1\}^\nu$ (i.e., the string $\tau_{\mathcal{A}}$). Hence, we can apply Lemma 10: this implies that there exists $\tau \in \{-1,1\}^\nu$ such that

$$\mathbb{E}\left[Q_T^{(\tau)}(L_{h_{\mathcal{A}}}) - Q_T^{(\tau)}(L_{h^*}^\tau)\right] \geq \frac{\Phi(\tilde{r})}{16\sqrt{6}} \min_{\substack{\tau',\tau'' \\ \mathrm{hd}(\tau',\tau'')=1}} \|Q^{(\tau')} \wedge Q^{(\tau'')}\|_1 .$$

We are left to evaluate the right-hand side of the above inequality. We can use the following known relations that hold for any two distributions $P$ and $Q$ over $\mathcal{Z}^T$ [Tsybakov, 2008]:

$$\|P \wedge Q\|_1 = 1 - \frac{1}{2}\|P - Q\|_1, \qquad \|P - Q\|_1 \leq \sqrt{2\mathrm{KL}(P,Q)} \tag{20}$$

where KL is the Kullback–Leibler divergence (we refer to the classic definition of those distances as in [Tsybakov, 2008]). Let $\tau'$ and $\tau''$ be two strings in $\{-1,1\}^\nu$ that only differ in one coordinate. Let $\mathrm{Ber}(p)$ be a Bernoulli distribution with parameter $p \in [0,1]$. We have that:

$$\begin{aligned}
\mathrm{KL}(Q^{(\tau')}, Q^{(\tau'')}) &= \sum_{t=1}^{T} \mathrm{KL}(Q_t^{(\tau')}, Q_t^{(\tau'')}) \\
&= \sum_{t=T-\tilde{r}+1}^{T} \mathrm{KL}(Q_t^{(\tau')}, Q_t^{(\tau'')}) \\
&= \frac{1}{\nu} \sum_{t=1}^{\tilde{r}} \mathrm{KL}\left(\mathrm{Ber}\left(\frac{1}{2} + \frac{1}{16\sqrt{6}}\left(\sqrt{\frac{\nu}{\tilde{r}}} + \Delta_t\right)\right), \mathrm{Ber}\left(\frac{1}{2} - \frac{1}{16\sqrt{6}}\left(\sqrt{\frac{\nu}{\tilde{r}}} + \Delta_t\right)\right)\right)
\end{aligned} \tag{21}$$

where the first equality is due to the factorization property of the KL-divergence, the second and third equality are due to the definition of the family $\mathcal{Q}$. For any $\epsilon < 1/4$, it holds that

$$\mathrm{KL}\left(\mathrm{Ber}\left(\frac{1}{2} - \epsilon\right), \mathrm{Ber}\left(\frac{1}{2} + \epsilon\right)\right) \leq 12\epsilon^2 .$$

By plugging the above inequality in (21), and using the fact that $(x+y)^2 \leq 2x^2 + 2y^2$, we obtain that

$$\begin{aligned}
\mathrm{KL}(Q^{(\tau')}, Q^{(\tau'')}) &\leq \frac{24}{16^2 \cdot 6\nu}\left[\sum_{t=1}^{\tilde{r}} \frac{\nu}{\tilde{r}} + \sum_{t=1}^{\tilde{r}} \Delta_{\tilde{r}}^2\right] \\
&\leq \frac{24}{16^2 \cdot 6\nu}\left[\nu + \tilde{r}\Delta_{\tilde{r}}^2\right]
\end{aligned}$$

By using the definition of $\tilde{r}$, it holds that $\Delta_{\tilde{r}}^2 \tilde{r} \leq \nu$. Thus, we we have that $\mathrm{KL}(Q^{(\tau')}, Q^{(\tau'')}) \leq 1/32$. If we use this inequality with (20), we have

$$\min_{\substack{\tau',\tau'' \\ \mathrm{hd}(\tau',\tau'')=1}} \|Q^{(\tau')} \wedge Q^{(\tau'')}\|_1 \geq 7/8 .$$

Hence, we can conclude that there exists $\tau \in \{-1,1\}^\nu$ such that

$$\mathbb{E}\left[Q_T^{(\tau)}(L_{h_{\mathcal{A}}}) - Q_T^{(\tau)}(L_{h^*}^\tau)\right] \geq \frac{7\sqrt{6}\Phi(\tilde{r})}{768} .$$

$\square$

By using this Lemma, we can easily prove Theorem 8.

*Proof of Theorem 8.* . By Lemma 12, there exists $\tau$ such that if $P_t = Q_t^{(\tau)}$ for all $1 \leq t \leq T$, then the algorithm $\mathcal{A}$ with input $Z_1, \ldots, Z_T$ satisfies

$$\mathbb{E}\left[P_T(L_{h_{\mathcal{A}}}) - P_T(L_{h^*})\right] \geq 7\sqrt{6}\Phi^*/768 . \tag{22}$$

Due to (18), we observe that the distributions $P_1, \ldots, P_T$ satisfy the assumption on the drift.

Let $E = P_T(L_{h_A}) - P_T(L_{h^*})$. Observe that due to the construction of $Q^{(\tau)}$ and (19), we have that $E \leq \Phi(\tilde{r})/(8\sqrt{6}) \leq 3\Phi^*/(8\sqrt{6})$, where the last inequality is due to Proposition 11. Let $\alpha > 0$ be a later defined value. We have that:

$$\mathbb{E}[E] = \mathbb{E}[E|E > \alpha\Phi^*]\Pr(E \geq \alpha\Phi^*) + \mathbb{E}[E|E < \alpha\Phi^*]\Pr(E \leq \alpha\Phi^*)$$
$$\leq (3\Phi^*/(8\sqrt{6}))\Pr(E \geq \alpha\Phi^*) + \alpha\Phi^*(1 - Pr(E \leq \alpha\Phi^*))$$
$$= \Pr(E \geq \alpha\Phi^*)[3\Phi^*/20 - \alpha\Phi^*] + \alpha\Phi^* .$$

By using the above inequality together with (22), we finally obtain that:

$$\Pr(E \geq \alpha\Phi^*) \geq \frac{7\sqrt{6}\Phi^*/768 - \alpha\Phi^*}{3\Phi^*/(8\sqrt{6}) - \alpha\Phi^*} = \frac{7\sqrt{6}/768 - \alpha}{3/(8\sqrt{6}) - \alpha}$$

We set $\alpha = 1/(112\sqrt{6})$ and we finally obtain that

$$\Pr\left(E \geq \frac{\Phi^*}{112\sqrt{6}}\right) \geq 1/8$$

.

$\square$

## B  Relaxing Assumption 2

Given a family $\mathcal{F}$, it is possible that the exact computation of $\|\mathbb{P}_T^r - \mathbb{P}_T^{2r}\|_\mathcal{F}$ is computationally hard. In this section, we show an example on how to relax Assumption 2 to allow an approximation of this quantity.

Let $\alpha \geq 1$. We say that an algorithm $A$ in an $\alpha$-approximation procedure if given $Z_{t-2r+1}, \ldots, Z_T$, it computes an estimate $A(Z_{T-2r+1}, \ldots, Z_T)$ such that

$$1 \leq \frac{\|\mathbb{P}_T^r - \mathbb{P}_T^{2r}\|_\mathcal{F}}{A(Z_{T-2r+1}, \ldots, Z_T)} \leq \alpha$$

for any $r \leq T/2$. That is, the algorithm $A$ does not compute the value of the supremum of the norm $\|\cdot\|_\mathcal{F}$ exactly, but it guarantees a constant factor approximation $\alpha$. In this case, we can modify the algorithm as follows.

---

**Algorithm 2:** Adaptive Learning Algorithm under Drift with Approximation Procedure

---

$i \leftarrow 0$ ;
**while** $r_i \leq T/2$ **do**
  **if** $A(Z_{t-2r+1}, \ldots, Z_T) \leq 4 \cdot S(r_i, \delta)$ **then**
    | $i \leftarrow i+1$ ;
  **end**
  **else**
    | **return** $r_i$ ;
  **end**
**end**
**return** $r_i$

---

**Theorem 13.** *Let $\delta \in (0, 1)$. Let Assumptions 1 hold, and assume that there exists an $\alpha$-approximation procedure for estimating $\|\mathbb{P}_T^r - \mathbb{P}_T^{2r}\|_\mathcal{F}$ for any $r \leq T/2$. Given $Z_1, \ldots, Z_T$, there exists an algorithm that outputs a value $\hat{r} \leq T$ such that with high-probability, it holds that*

$$\|P_T - \mathbb{P}_T^{\hat{r}}\|_\mathcal{F} = \alpha \cdot O\left(\min_{r \leq T}\left[\frac{\alpha C_{\mathcal{F},1}}{\sqrt{r}} + \max_{t < r}\|P_T - P_{T-t}\|_\mathcal{F} + \alpha C_{\mathcal{F},2}\sqrt{\frac{\log(\log(r+1)/\delta)}{r}}\right]\right)$$

*Proof.* The proof follows the same strategy as the one of the main theorem with slight modifications, and we discuss those changes. We replace the definition of $U(r, \delta)$ with

$$U(r, \delta) = (18\alpha + 9)S(r, \delta) + \|P_T - P_T^r\|_\mathcal{F}$$

in order to take into account the additional error due to the approximation procedure. We can observe that

$$A(Z_{T-r_{i+1}-1}, \ldots, Z_T) \leq 4S(r_i, \delta)$$

implies

$$\|\mathbb{P}_T^{r_i} - \mathbb{P}_T^{r_{i+1}}\|_{\mathcal{F}} \leq 4\alpha S(r_i, \delta)$$

since $A$ is an $\alpha$-approximation procedure. We can follow the steps of Proposition 4 with those different constants to show an equivalent statement of this Proposition. On the other hand, we have that if

$$A(Z_{t-2r+1}, \ldots, Z_T) > 4S(r_i, \delta)$$

then

$$\|\mathbb{P}_T^r - \mathbb{P}_T^{2r}\|_{\mathcal{F}} > 4S(r_i, \delta) \ ,$$

and Proposition 5 applies. Therefore, we can use the same proof strategy of Theorem 1 to prove Theorem 13

$\square$

