# OpenReview forum: "An Adaptive Algorithm for Learning with Unknown Distribution Drift"
_NeurIPS.cc/2023/Conference — NeurIPS 2023 poster_

### Official Review · Reviewer_ePCf · 2023-06-26

**Soundness:** 3 good
**Presentation:** 3 good
**Contribution:** 3 good
**Rating:** 4
**Confidence:** 3

**Summary:**

The paper proposes an algorithm for environments with changing distributions without assuming a priori knowledge about the change in distributions. The proposed algorithm provides error bounds that decrease with the number of time steps.

**Strengths:**

The idea of considering independent distributions with distribution shift is unexplored.

The algorithm itself is a solid technical contribution, and the theoretical results are strong


**Weaknesses:**

Numerical experiments are not included.

Assumptions need further justification.


**Questions:**

How strong is the assumption on the distributions? What scenarios satisfy such assumptions, are there any real examples?

The reviewer believes that the problem formulation needs further explanation (e.g., what is to be learned at each instant of time).

Numerical results could help to see the superiority of the presented method.

The references are a bit old. There are more recent related works:

“Minimax Classification under Concept Drift with Multidimensional Adaptation and Performance Guarantees.”

“Adaptive online learning in dynamic environment.”

“Random feature based online multi-kernel learning in environments with unknown dynamics.”

**Limitations:**

Numerical experiments are not included.

---

> ### Author Rebuttal · Authors · 2023-08-09
>
> Thank you for reviewing our paper and for your feedback.
>
> ---
>
> **Question**: The idea of considering independent distributions with distribution shift is unexplored.
>
> **Answer**: We study the classical setting where we have a sequence of drifting distributions, but the samples from those distributions are independent.
> As we discuss in the introduction, this idea is not unexplored, see e.g. [B,C] (additionally, references in lines 20-26). Our work improves upon the results of this line of research, solving an open question posed in [A] in this setting.
>
> ----
>
> **Question**:  Assumptions need further justification. How strong is the assumption on the distributions? What scenarios satisfy such assumptions, are there any real examples?
>
> **Answer**: This classical setting has been justified and studied in a long line of research, see lines 20-26 in the paper and references therein. The work [B] provides a good discussion of this setting. We are addressing an open theoretical question within this line of research [A], improving upon previous work.
>
> *In detail*:
> In Section 5, we discuss the assumptions introduced in the paper for the major problem of binary classification.
>
> For Assumption 2, we provide an additional discussion in lines 312-325 in the Conclusion, and also in Appendix B.
>
> As we describe in the paper, Assumption 1 is related to learnability, and we discuss sufficient conditions in lines 78-83 for it to hold. This is a very well-studied problem in the learning community, and indeed, these constants depend on the family $\mathcal{F}$, and a general discussion on how to obtain them is out of the scope of this paper.
>
> ----
>
> **Question** The reviewer believes that the problem formulation needs further explanation (e.g., what is to be learned at each instant of time).
>
> **Answer** The goal and the problem formulation are specified in Lines 60-62, and further discussed in the following lines. We want to estimate the expectation of all the functions in a family $\mathcal{F}$ with respect to the current distribution $P_T$. This formulation of learning and notation is common in the statistical learning/empirical process community (e.g., [D] ).
>
> ---
>
> **Question**: Numerical results could help to see the superiority of the presented method.
>
> **Answer**: The nature of our work is theoretical, and there is no other equivalent method to compare to. We improve upon a long line of research by solving an open question [A]: we show that we can obtain the same results of the previous work without knowing the drift a priori. Our results are also tight in a mini-max sense (e.g., Theorem 7 for binary classification).
>
> ----
>
> **Question**
> The references are a bit old.
> There are more recent related works:
>
> 1. “Minimax Classification under Concept Drift with Multidimensional Adaptation and Performance Guarantees.”
>
> 2. “Adaptive online learning in dynamic environment.”
>
> 3. “Random feature based online multi-kernel learning in environments with unknown dynamics.”
>
> **Answer**: Thank you for the additional references that we could add to the paper. We believe that we discuss the most relevant line of research in the introduction and related work. Our work solves an open question that was posed in a **recent** paper [A].
>
> The setting, the assumptions, and the main results of the mentioned papers are significantly different from our work. E.g:
>
> 1. The concept drift follows a specific dynamic described in that reference. Our paper addresses a more general setting.
>
> 2,3. Among the clearest differences, the measure of error is regret rather than the learning error for the distribution at the current time.
>
> ----
>
> [A]: Steve Hanneke and Liu Yang. Statistical learning under nonstationary mixing processes. AISTATS, 2019.
>
> [B]: Mohri, Mehryar, and Andres Muñoz Medina. "New analysis and algorithm for learning with drifting distributions." ALT, 2012.
>
> [C]:  Philip M Long. The complexity of learning according to two models of a drifting environment. COLT, 1998.
>
> [D]: Sen, Bodhisattva. "A gentle introduction to empirical process theory and applications." Lecture Notes, Columbia University 11 (2018): 28-29.

---

> > ### Comment · Reviewer_ePCf · 2023-08-19
> >
> > Thank you for the detailed rebuttal. You have addressed all my questions/concerns
> > After reading the rebuttal and other reviews, my recommendation remains the same.

---

### Official Review · Reviewer_k7dA · 2023-07-05

**Soundness:** 3 good
**Presentation:** 3 good
**Contribution:** 3 good
**Rating:** 4
**Confidence:** 3

**Summary:**

This paper under considers the theoretical problem of determining the sliding window size for empirical risk minimization in the presence of unknown distribution changes. The proposed method aims to enable the learning of a classifier comparable to approaches that have prior knowledge of the distribution change magnitude.

**Strengths:**

+ The paper's motivation is commendable, as it aims to tackle the challenge of non-stationary sequential learning, specifically with unknown distribution shifts. This is an important problem, and the proposed technique could potentially contribute to its advancement.

+ Although not evaluated extensively, the results appear reasonable. While I haven't reviewed the proofs in detail, they seem sound.

**Weaknesses:**


- The paper relies on computability assumptions for key variables, such as the distribution discrepancy (e.g., $\|P^r_T - P^r_T\|_{\mathcal{F}}$) and the constants ($C_{\mathcal{F},1}$ and $C_{\mathcal{F},2}$). However, the paper lacks a discussion on when these assumptions hold, how solvable they are, or how accurately they can be estimated or upper bounded. In comparison to previous methods that estimate distribution shifts, it is unclear whether estimating these parameters would lead to a more practical algorithm. This aspect requires further analysis and discussion.

- The authors claim that their proposed algorithm aims to provide a practical solution to handling unknown sequential distribution shifts without prior knowledge. However, the proposed algorithm introduces new parameters that are challenging to estimate or even tightly upper bound, such as the distribution discrepancy (assumption 2). While the new analysis techniques presented may contribute to addressing the problem, the overall contribution seems limited, as the proposed method essentially shifts the difficulty from estimating the distribution shift magnitude to estimating other parameters.

- The paper lacks a thorough discussion on when Assumption 2 holds, especially in practical scenarios. This assumption is crucial, as the proposed method relies on estimating the distribution discrepancy and other constants that evolve the hypothesis class $\mathcal{F}$. Without addressing the practicality of these estimations, the proposed method's feasibility remains questionable. It would be beneficial to provide a detailed comparison of the estimation difficulty for the parameters required in the proposed method compared to previous works.

**Questions:**

It is important to explicitly discuss the computability assumptions made in the paper and discuss when and how they hold. This discussion should include considerations of solvability, accurate estimation, or upper bounding of variables such as the distribution discrepancy and the constants. So it is suggested to provide an in-depth analysis of Assumption 2 and other computability assumptions on the constants mentioned above and its applicability in practical scenarios.

**Limitations:**

Limitations are discussed in this work.

---

> ### Author Rebuttal · Authors · 2023-08-09
>
> Thank you for your work in reviewing our manuscript, and for providing thorough feedback.
>
> ---
>
> **Question**: The paper relies on computability assumptions for key variables, such as the distribution discrepancy (e.g., $|P^r_T - P^r_T|_{\mathcal{F}},
> C{\mathcal{F},1}
> C_{\mathcal{F},2}$). However, the paper lacks a discussion on when these assumptions hold, how solvable they are, or how accurately they can be estimated or upper-bounded. In comparison to previous methods that estimate distribution shifts, it is unclear whether estimating these parameters would lead to a more practical algorithm. This aspect requires further analysis and discussion.
>
> **Question**: The paper lacks a thorough discussion on when Assumption 2 holds, especially in practical scenarios. This assumption is crucial, as the proposed method relies on estimating the distribution discrepancy and other constants that evolve the hypothesis class
> $\mathcal{F}$. Without addressing the practicality of these estimations, the proposed method's feasibility remains questionable [...].
>
> **Answer**: Section 5 gives a thorough discussion of both Assumptions 1 and Assumptions 2 for the important case of binary classification.
>
> For Assumption 2, we provide an additional discussion in lines 312-325 in the Conclusion, and also in appendix B.
>
> As we describe in the paper, Assumption 1 is related to learnability, and we discuss sufficient conditions in lines 78-83 for it to hold. This is a very well-studied problem in the statistical learning community, and indeed, these constants depend on the family F. A general discussion on how to obtain them is out of the scope of this paper.
>
> ----
>
> **Question**: The authors claim that their proposed algorithm aims to provide a practical solution to handling unknown sequential distribution shifts without prior knowledge. However, the proposed algorithm introduces new parameters that are challenging to estimate or even tightly upper bound, such as the distribution discrepancy (assumption 2). While the new analysis techniques presented may contribute to addressing the problem, the overall contribution seems limited, as the proposed method essentially shifts the difficulty from estimating the distribution shift magnitude to estimating other parameters.
>
> **Answer**:  We respectfully disagree with this characterization, because previous work required prior knowledge, whereas in our work we obtain similar results while only relying on quantities that can be estimated from the data. Moreover, we completely characterize those assumptions for the major problem of binary classification, improving upon a long sequence of work in that area (Section 5). In the general case, the assumptions depend on $\mathcal{F}$.
>
> We also remark that in this setting it is not possible to estimate the magnitude of the distribution drift since there is only one sample from each distribution.
>
> *More details*:
> The constants $C_{\mathcal{F},1}$ and $C_{\mathcal{F},2}$ are related to the complexity of learning the family $\mathcal{F}$. This is a well-studied problem within the statistical learning community, and these constants depend on the family $\mathcal{F}$.
>
> The hardness of evaluating the discrepancy in the general case is not a limitation unique to our work, and it is a known limitation for work on transfer learning/domain adaptation, as we remark in the Conclusion. In fact, even with access to more samples from each distribution, previous work has the challenge of estimating the discrepancy (e.g., [C,D]). In our case, we show a major setting (binary setting) in which it is efficiently computable even with limited data.  There are other cases for which the computation of the discrepancy is known to be possible [B] (e.g., regression with squared loss)
>
> ----
>
> References:
>
> [B]: Mansour, Yishay, Mehryar Mohri, and Afshin Rostamizadeh. "Domain adaptation: Learning bounds and algorithms." COLT, 2009
>
> [C]: Mohri, Mehryar, and Andres Muñoz Medina. "New analysis and algorithm for learning with drifting distributions." ALT, 2012.
>
> [D]: Awasthi, Pranjal, Corinna Cortes, and Christopher Mohri. "Theory and algorithm for batch distribution drift problems." AISTATS, 2023.

---

> > ### Comment · Reviewer_k7dA · 2023-08-19
> >
> > After checking the rebuttal and other reviewers' comments, I find that the response does not adequately address my concerns. Main weaknesses still exists:
> >
> > 1. The proposed theorem requires an accurate estimation of the distribution discrepancy, which is also related to the hypothesis classes. This is hard to estimate and verify in real applications and thus limits the application scope of this work.
> > 2. The authors claim that "We respectfully disagree with this characterization, because previous work required prior knowledge, whereas in our work we obtain similar results while only relying on quantities that can be estimated from the data." From a theoretical point of view, it seems to be overclaimed, see [1,2] and the following papers and references therein (It is also suggested to include the discussion with this line of works in the main paper). In my opintion, the new analysis techniques proposed in this draft may contribute to addressing the real-world problem associated with general model and loss functions, but the theoretical results are still rather preliminary and overall contribution and seems limited. Also, there are no experiments.
> >
> > Therefore, I tend to remain my sorce.
> >
> > [1] Cutkosky, A.. Parameter-free, dynamic, and strongly-adaptive online learning. In ICML, pp. 2250-2259, 2020.
> >
> > [2] Wei, C. Y., & Luo, H. Non-stationary reinforcement learning without prior knowledge: An optimal black-box approach. In COLT, pp. 4300-4354, 2021.

---

> > > ### Author Response · Authors · 2023-08-21
> > > **Response**
> > >
> > > We thank the reviewer for their throughout feedback, and for providing a detailed response to our rebuttal.
> > >
> > > We would like to provide a response to the above points.
> > >
> > > ----
> > >
> > > Previous work (2): "From a theoretical point of view, it seems to be overclaimed [...]"
> > >
> > > We thank the reviewer for providing these additional references. While indeed these papers are addressing a similar problem as they have distribution drift, the setting is clearly different. In these papers, the goal is to minimize the regret.
> > >
> > > Our claim is with respect to the different problem that is addressed in our paper: we focus on the best possible learning error at the current given time given the past observations.  We follow the line of research discussed in the paper (e.g., [2,4])  and we provide an answer to the **open problem** of the paper in [4] (see the end of Section 3 of that paper): how to find a learning algorithm that is adaptive (based on the data) with respect to the drift and theoretically as good as algorithms that known the drift a priori.
> > >
> > > ----
> > >
> > > Computability of discrepancy (1)
> > >
> > > a) While indeed it is true that the estimation is hard in the general case, we characterize this estimation and assumption for a binary family of functions (including binary classification). We devote a whole section (Section 5) to this setting. This is a major setting in Machine Learning. We also show that the output of our algorithm is tight in a minimax sense in this setting.
> > >
> > > b) We remark that the notion of discrepancy is not our contribution, it was introduced in previous work [1], and it has been used in the previous research mentioned in the paper and in our rebuttal. The discrepancy can be statistically estimated by the data, but its computation can be hard in the general case, indeed this is not a limitation unique to our work on drift [2,3], see in particular discussion in [2].
> > >
> > > c) While we could have only focused on the binary setting, we think that the presentation as $F$ being a general family of functions is a contribution, as it allows the trivial application of our algorithm to other settings. Indeed, there are other important families for which it is easy to compute the discrepancy. Examples:
> > > - if the domain $X$ is the real line, and $F$ contains the identity function only, this is a mean estimation problem;
> > > - if $F$ contains an indicator function for each possible subset of elements in a finite domain, this is a discrete density estimation (under total variation distance).
> > >
> > > Both examples point to other important learning problems in which the discrepancy is trivial to compute and easily fit within our framework.
> > >
> > > ----
> > >
> > > [1]: Mansour, Yishay, Mehryar Mohri, and Afshin Rostamizadeh. "Domain adaptation: Learning bounds and algorithms." COLT, 2009
> > >
> > > [2]: Mohri, Mehryar, and Andres Muñoz Medina. "New analysis and algorithm for learning with drifting distributions." ALT, 2012.
> > >
> > > [3]: Awasthi, Pranjal, Corinna Cortes, and Christopher Mohri. "Theory and algorithm for batch distribution drift problems." AISTATS, 2023.
> > >
> > > [4]: Steve Hanneke and Liu Yang. Statistical learning under nonstationary mixing processes. AISTATS, 2019.

---

> > > > ### Comment · Reviewer_k7dA · 2023-08-22
> > > >
> > > > Thank you for the detailed clarification. It is suggested that these discussions be included in the main paper.

---

### Official Review · Reviewer_RWUF · 2023-07-06

**Soundness:** 3 good
**Presentation:** 2 fair
**Contribution:** 2 fair
**Rating:** 4
**Confidence:** 3

**Summary:**

The author propose a general algorithm to learn a family of functions with respect to the current distribution at time T. This algorithm achieve a drifting-instance-dependent bound without any prior knowledge of the drift. Based on this, the author further analyze a tractable algorithm on binary classifier.

**Strengths:**

1. The paper solves an open problem of drifting distribution with unknown prior. The proof technique is clear and solid.
2. It's further analysis on binary classification is inspiring.

**Weaknesses:**

1. The error bounded is only measured on windows that end at T. Instead, a better goal would be to select a window from t1 to t2. As an example of why this problem could be more relevant, there might be large drifts nearby T but early distributions are close to P_T.

2. The techniques used in this paper is not very novel. Specifically, although the authors claim they do not directly estimate the drift error, it is still drift detection only with tolerance towards estimation error. The doubling trick is intuitive and not surprising here.

3. As the authors admitted in our paper, this algorithm is very general and might be computationally intractable in more complicated cases. I think the author could discuss the benefit of their algorithm versus existing work on cases beyond binary classification. They may do this by showing either further proofs or empirical evidence. Currently, it is not immediate to me the value of an algorithm that ignores assumptions on the magnitude of the drifts.

**Questions:**

Why is empirical estimate based on a window up to T better than empirical estimates based on any consecutive window?
What's the value in getting rid of prior knowledge on drift magnitude in practice or in theory?

**Limitations:**

Sufficient.

---

> ### Author Rebuttal · Authors · 2023-08-08
>
> Thank you for your close reading of our paper and for your feedback.
>
> ---
>
> **Question**: The error bounded is only measured on windows that end at T. Instead, a better goal would be to select a window from t1 to t2. As an example of why this problem could be more relevant, there might be large drifts nearby T but early distributions are close to P_T.
>
> **Question**: Why is empirical estimates based on a window up to T better than empirical estimates based on any consecutive window?
>
> **Answer**: One can modify our algorithm and analysis to handle intervals ending at any point in the past. However, without prior knowledge (such as periodicity) the most reasonable assumption is that observations that are closest in time provide the most relevant information. For this reason, a long line of research in this area considers windows that end at T (e.g., references in lines 20-26 in the paper); we improve upon this line of work, solving the open question in [A] for this setting.
>
> We remark that a properly chosen window that ends at T  provides a minimax tight estimation under simple assumptions (e.g., bounded drift at each step), see line 24, line 247, and Theorem 7.
>
> ----
>
> **Question**: The techniques used in this paper is not very novel. Specifically, although the authors claim they do not directly estimate the drift error, it is still drift detection only with tolerance towards estimation error. The doubling trick is intuitive and not surprising here.
>
> **Answer**: We respectfully disagree. We present the first algorithm that does not rely on any prior knowledge of the drift but only uses empirical measures learned from the data. We think this is a major contribution that solves an open problem posed in [A]. Obviously, the widely popular doubling trick is not a contribution of this paper.
>
> Our question to the reviewer is why do they not think this is not a novel contribution?
>
> ----
>
> **Question**: As the authors admitted in our paper, this algorithm is very general and might be computationally intractable in more complicated cases. I think the author could discuss the benefit of their algorithm versus existing work on cases beyond binary classification. They may do this by showing either further proof or empirical evidence. Currently, it is not immediate to me the value of an algorithm that ignores assumptions on the magnitude of the drifts.
>
> **Answer**: Binary classification is a major problem in machine learning. We think that even just this result  (Section 5) is a major contribution, as previous existing work relied on a priori assumption on the drift. In the general case, the hardness depends on the family F (as in almost all ML applications).  There are other cases for which the computation of the discrepancy is known to be possible [B] (e.g., regression with squared loss).  See also our discussion in lines 312-325
>
> The hardness of evaluating the discrepancy is not limited to our technique, and it is a known limitation for work on transfer learning/distribution drift. In fact, even with access to more samples from each distribution, previous work has the challenge of estimating the discrepancy (e.g., [C,D]). In our case, we show a major setting (binary setting) in which it is efficiently computable even with limited data.
>
> ----
>
> **Question**: What's the value in getting rid of prior knowledge on drift magnitude in practice or in theory?
>
> **Answer**: We eliminated the requirement of prior knowledge on the drift, providing similar guarantees without this knowledge and only relying on the data. This addresses an open problem in [A] prompted by the following reason: prior knowledge of the drift is unrealistic in practice.
>
> ----
>
> References:
>
> [A]: Steve Hanneke and Liu Yang. Statistical learning under nonstationary mixing processes. AISTATS, 2019.
>
> [B]: Mansour, Yishay, Mehryar Mohri, and Afshin Rostamizadeh. "Domain adaptation: Learning bounds and algorithms." COLT, 2009.
>
> [C]: Mohri, Mehryar, and Andres Muñoz Medina. "New analysis and algorithm for learning with drifting distributions." ALT, 2012.
>
> [D]: Awasthi, Pranjal, Corinna Cortes, and Christopher Mohri. "Theory and algorithm for batch distribution drift problems." AISTATS, 2023.

---

### Official Review · Reviewer_sYaX · 2023-07-07

**Soundness:** 3 good
**Presentation:** 3 good
**Contribution:** 3 good
**Rating:** 7
**Confidence:** 3

**Summary:**

This research paper presents a straightforward algorithm designed to facilitate adaptive learning of models in the presence of distribution drift. The algorithm is specifically designed to adapt to changing data patterns without requiring any prior knowledge of the drift. Moreover, the paper provides a proven bound that guarantees the learning error of the algorithm. Overall, the paper is well-written. The simplicity of the algorithm contributes to its comprehensibility, making the results easily understandable. Additionally, the inclusion of a detailed illustration showcasing the algorithm's effectiveness in handling binary classification with distribution drift serves as a strong validation of the proposed methodology.

I think this is a breakthrough work for learning under non i.i.d data as it does not require prior knowledge of the potential drift. However, this could also be one of the limits of this paper. The algorithm proposed is to adaptively find r that can achieve the best trade-off between the statistical and drift error. Drift sometimes might represent an upcoming arrival of a new distribution. The proposed algorithm might get a competitive average accuracy compared to those algorithms that know the magnitude of drift in advance. However, I am not sure if this proposed algorithm will perform equivalent good if we test it on every sequential portion of the data as well. For example, a classifier of labeling every instance as 0 could also achieve an accuracy of 90% on the data where 90% of instances are 0 class. In this case, I get very limited information from its good results.

Overall, I think the proposed methodology provides a new way of designing the learning algorithms under drift.

**Strengths:**

Overall, I think the proposed methodology provides a new way of designing the learning algorithms under drift.
I also agree that r would be an important factor during learning under drift. Therefore, adding this to the learning boundary, from my perspective, is a good start of conducting the following theoretical learning works that discuss the non-identically distributed problem. Theorem 1 could be a quite useful guidance for the following studies in this area.

**Weaknesses:**

As I have mentioned in the summary, Theorem 1 only provides a very general bound. The r is considered as sequentially increased. However, I think the usage of Theorem 1 is very limited if we apply it to learning sequential data. As for the sequential data, there should be always new data coming and thus very possibly new distribution in the upcoming instances. It is less practical to design the r as monotonically increasing if you don't consider the magnitude of the drift. The Inequality in Algorithm 1 is always comparing P_t between it is at r_i and r_{i+1}. This means the i is only updated when the distribution drift is large enough among two consecutive i. However, this is not true. Therefore, this algorithm removes the assumption of knowing the magnitude of distribution drift in advance, but for me, it actually adds a hidden assumption that the magnitude of distribution drift should be enough big between at least one of two consecutive i in [1, T].

Let's think of the situation that the magnitude of distribution drift is incremental increasing by i. I am not sure if your algorithm can output the expected r in such a case. But for those algorithms that assume a known magnitude of drift in advance. They can complete this task.

As is also mentioned in this paper that an adaptive algorithm with respect to the drift that uses distribution-dependent upper bounds

**Questions:**

In Theorem 1, r is the output of Algorithm 1. What will be the limits if using other algorithms?


**Limitations:**

Some of the limitations have been listed in section 7. As is discussed, satisfying Assumption 2 is challenging and authors already discuss possible solutions.

From my perspective, the author claims their contributions as removing the prior knowledge of distribution drift. I appreciate their contribution. But as a reader, I am more curious if we do get some prior knowledge of the distribution drift, how can we use that knowledge to improve what you have presented in this work? I think this will provide more insights of this work.

---

> ### Author Rebuttal · Authors · 2023-08-08
>
>
> Thank you for taking the time to review our paper and for your feedback.
>
> -----
>
> **Question**:  "The algorithm proposed is to adaptively find r that can achieve the best trade-off between the statistical and drift error. The proposed algorithm might get a competitive average accuracy compared to those algorithms that know the magnitude of drift in advance. However, I am not sure if this proposed algorithm will perform equivalent good if we test it on every sequential portion of the data as well. For example, a classifier of labeling every instance as 0 could also achieve an accuracy of 90% on the data where 90% of instances are 0 class. In this case, I get very limited information from its good results."
>
> **Answer**: Indeed, this is the first algorithm that doesn’t rely on prior knowledge about the drift.  The algorithm provides an optimal solution for the aforementioned trade-off for any given time T (Theorem 1) (not an average over T).  Its solution is provably as tight as the one of other algorithms that require knowing the magnitude of the drift in advance, solving the open question in [A].
>
> ----
>
> **Question**: "[...] The Inequality in Algorithm 1 is always comparing P_t between it is at r_i and r_{i+1}. This means the i is only updated when the distribution drift is large enough among two consecutive i. However, this is not true. Therefore, this algorithm removes the assumption of knowing the magnitude of distribution drift in advance, but for me, it actually adds a hidden assumption that the magnitude of distribution drift should be enough big between at least one of two consecutive i in [1, T]"
>
> **Question**: "Let's think of the situation that the magnitude of distribution drift is incremental increasing by i. I am not sure if your algorithm can output the expected r in such a case. But for those algorithms that assume a known magnitude of drift in advance. They can complete this task.
> It is not true that we require the drift to be big enough between two consecutive i in [1,T]."
>
> **Answer**: r_i and r_{i+1}  are not two consecutive time steps.  r_{i+1} = 2r_i  (see line 156). As i increases, this gap is sufficient to detect even small gradual drift, as also discussed in the example in lines 241-256 in the paper.
>
> ----
>
> **Question**: In Theorem 1, r is the output of Algorithm 1. What will be the limits if using other algorithms?
>
> **Answer**: Algorithm 1 provides an optimal solution of the trade-off, as proven in Theorem 1, which is tight up to constants for binary classification (Theorem 7, minimax). There are no other algorithms to compare to since we are addressing an open question, and there are no other algorithms that provide such guarantees.
>
> ----
>
> **Question**: From my perspective, the author claims their contributions as removing the prior knowledge of distribution drift. I appreciate their contribution. But as a reader, I am more curious if we do get some prior knowledge of the distribution drift, how can we use that knowledge to improve what you have presented in this work? I think this will provide more insights of this work.
>
> **Answer**: The goal of our work is to eliminate the requirement of prior knowledge of the drift, providing similar guarantees without this knowledge. This prior knowledge is unrealistic in practice, and this is the main reason that prompts the open question of [A].
>
> ----
>
> References.
> [A]: Steve Hanneke and Liu Yang. Statistical learning under nonstationary mixing processes. In The 22nd International Conference on Artificial Intelligence and Statistics, pages 1678–1686. PMLR, 2019

---

> > ### Comment · Reviewer_sYaX · 2023-08-14
> >
> > Thanks for your replies.
> > "r_i and r_{i+1} are not two consecutive time steps. r_{i+1} = 2r_i (see line 156). As i increases, this gap is sufficient to detect even small gradual drift"
> >
> > By saying "Let's think of the situation that the magnitude of distribution drift is incremental increasing by i. I am not sure if your algorithm can output the expected r in such a case. But for those algorithms that assume a known magnitude of drift in advance. They can complete this task. It is not true that we require the drift to be big enough between two consecutive i in [1,T].",
> > I didn't mean your method cannot detect the drift. I was questioning if your algorithm can output the expected r. As I doubted, "it is less practical to design the r as monotonically increasing", i here increases fast by iterations. If a drift is detected given a large i, does it mean this detection is largely delayed compared to its ground truth when this drift just occurs?

---

> > > ### Author Response · Authors · 2023-08-14
> > > **Reply to Comment**
> > >
> > > Thank you for your quick reply and for clarifying this question.
> > >
> > > In our work, we do not explicitly address a drift detection problem, as our goal is to provide an optimal learning with respect to the current distribution. The drift detection problem is not trivial to define in our setting, since each distribution could be different.
> > >
> > > Indeed, there is a window size $r^*$ that optimally minimizes the trade-off between statistical error and drift error in the estimation.
> > > Our algorithm provides a window size $\hat{r}$ that yields an estimation error which is up to constants as good as the minimum given by $r^*$ [Theorem 1]. That is, the learning error of our algorithm is as good as if we had access to the unknown ground-truth about the drift, and we could compute $r^*$. This is a formal guarantee of the algorithm [Theorem 1], and we do not have any restrictions on the drift: this is the case for both an abrupt drift or a gradual drift. Note that we do not claim that $\hat{r} \simeq r^*$, and it is possible that $\hat{r}$ is possibly smaller or larger than $r^*$.

---

> > > > ### Comment · Reviewer_sYaX · 2023-08-15
> > > >
> > > > yes, I shouldn't have this question. Thanks.

---

### Decision · Program_Chairs · 2023-09-21

**Decision:**

Accept (poster)

**Comment:**

This paper considers the problem of learning under distribution shift. More generally consider the problem of learning from n samples where each sample $x_i$ is an i.i.d draw from a distribution $P_i$. Furthermore it is known that any two consecutive distributions are $\Delta$-close in TV distance. If $\Delta$ is known before hand then the best learning guarantee on can obtain is O((\Delta OPT)^{1/3}) where OPT is the error of the best function in class. It was open as to whether one can obtain the same guarantee without the knowledge of $\Delta$. The current paper solves this problem under the assumption that the labeled Y-discrepancy can be solved efficiently. The approach is simple but neat. It involves adaptively estimating the right sample size by repeatedly invoking an oracle for solving the discrepancy problem over samples in a window of size r and 2r for various values of r.

One reviewer very enthusiastically recommended the acceptance of the paper whereas the rest did not increase their scores during the discussion phase. The main points raised were: a) moving the difficulty of the problem from knowing $\Delta$ to solving a hard optimization problem, b) the "doubling" trick being not novel, c) lack of experiments.

Based on my own independent reading of the paper point c) is irrelevant. This is a theory paper and should be judged accordingly. Point b) is also unfair since the way the doubling idea is applied is clever in my opinion. Finally, I feel that point a) is not that serious of a problem since in many settings such as classification and linear regression, estimating discrepancy boils down to invoking an ERM oracle which is a very reasonable oracle to assume access to.

In my opinion this paper presents an important and novel advancement to the problem of learning under unknown drift and would be a very good addition to NeurIPS.